# Increasing species sampling in chelicerate genomic-scale datasets provides support for monophyly of Acari and Arachnida

Jesus Lozano-Fernandez [1,2,5,6], Alastair R. Tanner[1,6], Mattia Giacomelli[1], Robert Carton [3], Jakob Vinther [1,2], Gregory D. Edgecombe[4] & Davide Pisani [1,2]

Chelicerates are a diverse group of arthropods, represented by such forms as predatory spiders and scorpions, parasitic ticks, humic detritivores, and marine sea spiders (pycnogonids) and horseshoe crabs. Conflicting phylogenetic relationships have been proposed for chelicerates based on both morphological and molecular data, the latter usually not recovering arachnids as a clade and instead finding horseshoe crabs nested inside terrestrial Arachnida. Here, using genomic-scale datasets and analyses optimised for countering systematic error, we find strong support for monophyletic Acari (ticks and mites), which when considered as a single group represent the most biodiverse chelicerate lineage. In addition, our analysis recovers marine forms (sea spiders and horseshoe crabs) as the successive sister groups of a monophyletic lineage of terrestrial arachnids, suggesting a single colonisation of land within Chelicerata and the absence of wholly secondarily marine arachnid orders.

[1] University of Bristol School of Biological Sciences, 24 Tyndall Avenue, Bristol BS8 1TQ, UK. [2] University of Bristol School of Earth Sciences, 24 Tyndall Avenue, Bristol BS8 1TQ, UK. [3] Department of Biology, The National University of Ireland Maynooth, Maynooth, Kildare, Ireland. [4] Department of Earth Sciences, The Natural History Museum, Cromwell Road, London SW7 5BD, UK. [5] Present address: Department of Evolutionary Biology, Ecology and Environmental Sciences, & Biodiversity Research Institute (IRBio) Universitat de Barcelona, Avinguda Diagonal 643, Barcelona 08028, Spain. [6] These authors contributed equally: Jesus Lozano-Fernandez, Alastair R. Tanner. Correspondence and requests for materials should be addressed to G.D.E. (email: g.edgecombe@nhm.ac.uk) or to D.P. (email: davide.pisani@bristol.ac.uk)

Chelicerata is the second largest subphylum of arthropods, outnumbered only by insects. They exhibit enormous terrestrial diversity, including web-building predatory spiders, parasites such as hemophage ticks, mites (some of which are subsocial), amblypygids (whip scorpions), opiliones (harvestmen), and ricinuleids (hooded tick-spiders), as well as having marine representatives: the horseshoe crabs and pycnogonids (sea spiders). The evolutionary history of the chelicerates extends back at least to the Cambrian, around 524 million years ago (Ma), as can be inferred from both the fossil record and molecular divergence time estimation[1–3]. As predatory components of diverse ecosystems, the rock record shows that chelicerates have been key in both earlier Palaeozoic marine settings[4], and later Mesozoic and Cenozoic[5,6] marine and terrestrial ecosystems. Most chelicerate diversity is represented by the Arachnida, a lineage traditionally assumed to be exclusively terrestrial, composed of 12 groups classified as orders, each of undisputed monophyly[7]. However, the relationships among these lineages continue to be debated, with substantial discrepancies between alternative morphological and molecular datasets. Recent phylogenomic analyses achieved some progress by reducing methodological biases, increasing phylogenetic signal, and applying better fitting substitution models[8,9]. These studies recovered several established high-level groups (Chelicerata, Euchelicerata, and Tetrapulmonata) and intriguingly proposed that the horseshoe crabs might be a secondarily marine, arachnid lineage[9]. Recent studies were unable to provide firm conclusions for the interrelationships between mites and ticks[10,11], traditionally considered to represent a monophyletic group (Acari). However, morphological support for Acari is not unanimous and the two subgroups have also been suggested to constitute independently evolved lineages: Acariformes and Parasitiformes[10–12], including the medically important haemophages.

We present a phylogenomic investigation of Chelicerata, utilising both new and more complete sequence information and a robust inferential methodology. Our results indicate that Acari constitutes a monophyletic group nested within a monophyletic Arachnida. With more than 55,000 described species[13], Acari is thus the most biodiverse chelicerate clade. In addition, our results suggest that the marine chelicerates (sea spiders and horseshoe crabs) are the successive sister groups of the terrestrial Arachnida, consistent with a single, unreversed, colonisation of land underpinning the evolutionary success of this group.

## Results and Discussion

**Taxonomic sampling**. We compiled molecular datasets based on transcriptomic data from 95 species (Supplementary Table 1), four of which are newly sequenced. Our dataset includes representatives of all arachnid orders, with the exception of two minor lineages, Palpigradi and Schizomida (the latter being widely accepted as the sister lineage of Uropygi[14]), for which genomic-scale data are still missing. We include sequence information from three of four living horseshoe crab species, and most importantly we expand taxon sampling within mites and ticks, including short-branched representatives of Sarcoptiformes (Acariformes) and Mesostigmata (Parasitiformes). The inclusion of short-branched Acariformes and Parasitiformes is of particular relevance because previous studies suggested that the chelicerate tree is prone to the effects of Long Branch Attraction (LBA) artifacts[8,15], and long-branched mites and ticks are particularly common across previous datasets[8].

**Matrix assembly and model selection**. To test chelicerate relationships, we generated three datasets using two alternative orthology selection strategies. For the first dataset (Matrix A) we

expanded our legacy dataset[2,16,17] to generate a superalignment including 45,939 amino acid positions (78.1% complete) from 233 loci. The genes in Matrix A were originally selected to maximise inclusion of known single-copy genes (to minimise the negative effects of hidden paralogy) as well as slowly evolving genes (to reduce the negative impact of saturation-dependent tree reconstruction artifacts, like LBA[18]). Matrix A is thus enriched in informational genes[19], e.g. ribosomal proteins, that are frequently present in single copy and are characterised by a low rate of evolution[20]. For our second dataset (Matrix B) we used the OMA stand-alone software[21] to de novo identify orthologs from our set of transcriptomes. Using OMA we generated a new set of 3982 loci based on retaining genes for which orthologs were present in at least in 35% or more of the 95 considered taxa. Each of these high occupancy loci was then concatenated into a final supermatrix that was trimmed to remove poorly aligned regions. The final, trimmed, supermatrix included 34,839 amino acid positions (86.4% complete); see Methods for details. The third dataset (Matrix C) was created by concatenating all genes from Matrix B that are not in Matrix A (see Supplementary Table 2 and Methods). Matrix C was then trimmed using the same software and parameters used for Matrix B. While Matrices A and B are not fully independent datasets, Matrices A and C (the latter including 30,552 amino acids) are, and Matrix C was assembled to better establish the extent to which similarities between the trees derived from Matrices A and B might have been caused by the presence of a common set of shared sites (Supplementary Table 2). In light of previous suggestions that chelicerate trees might be prone to LBA artifacts[8,15], we used saturation plots to compare substitutional-saturation levels in these three matrices, as an increased level of saturation is a factor that can lead to the recovery of tree reconstruction artifacts[18,22,23]. Results of saturation-plot analyses provide us with an objective way to rank a-priori the expected relative phylogenetic utility of our datasets. The saturation plots (Fig. 1a) indicated that Matrix A, which was originally designed to exclude fast evolving genes, is less saturated ($R^2 = 0.74$) than are Matrices B and C ($R^2 = 0.42$ and 0.48, respectively). These latter two are in effect equally saturated: they display minimal differences in $R^2$ values and regression lines of almost identical slope (Fig. 1a). These results are not unsurprising given that Matrices B and C were pooled from a very large set of orthologs that was not filtered to remove genes with a high rate of evolution, and Matrix C only differs from Matrix B in excluding a small (9.33%), random sampling of sites that are also present in Matrix A. Overall, these results indicate that while the strategy used to derive Matrices B and C allows homogeneously sampling genes from across genomes, it retains a greater number of substitutionally-saturated genes, and according to the considered criterion Matrix A should be considered the most reliable dataset. Matrices B and C were subjected to very stringent trimming protocols (to minimise missing data) implemented using the software Gblocks[24] (see Methods for details). We performed a sensitivity analysis to test whether results of the analyses of Matrix B and C might have been biased by our site trimming strategy. To do so, we generated a fourth dataset (Matrix D) applying less stringent trimming criteria, and using a different alignment trimming software (trimAl[25]), to the untrimmed version of Matrix B. Matrix D includes 127,114 amino acids and is more than three times larger than Matrix B.

We performed model testing (using Bayesian Cross Validation[26] on Matrix A) and found that the compositionally site-heterogeneous CAT-GTR + G model fits the data best: Cross-Validation Score = 21.1139 ± 8.39011 (in favour of CAT-GTR + G—against the second best fitting model GTR + G). This result was expected as CAT-GTR + G[27] has been shown to possess a greater ability to adequately describe site-specific amino acid

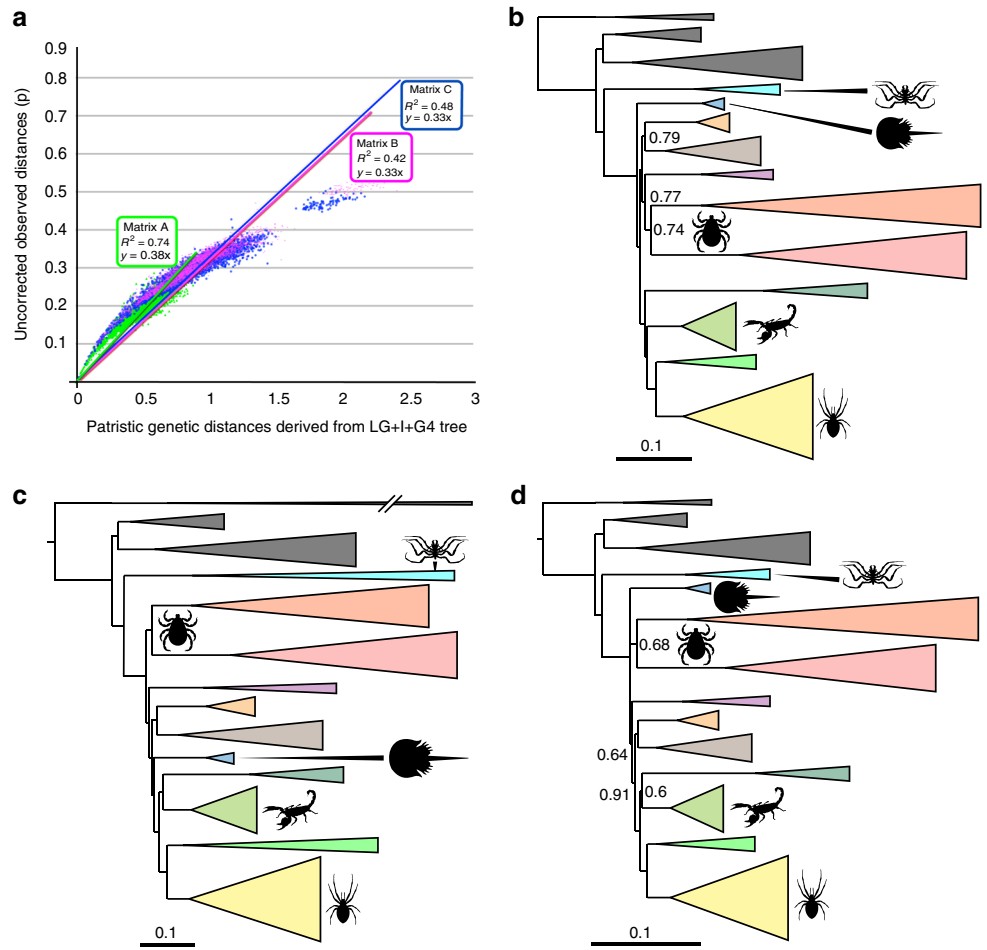

**Fig. 1** Saturation plots and Bayesian results. **a** Saturation plots for Matrices A, B and C, illustrating that Matrix A has the lowest level of saturation. The dots represent the intersection between uncorrected genetic distances and the patristic distances obtained under ML for each pair of the 95 taxa. The lines of best fit show the general trend and direction of the data and above is shown the $R^2$ and the slope of the regression line. Matrix A is represented in green, Matrix B in magenta and Matrix C in blue. Source data are provided as a Source Data file. **b** Schematic representation of the results of the CAT-GTR + G analysis of Matrix A. **c** Schematic representation of the results of the CAT-GTR + G analysis of Matrix B. **d** Schematic representation of the results of the CAT-GTR + G analysis of Matrix A, after Dayhoff-6 recoding. **b**–**d** Lineages are summarized at the ordinal level. Support values represent posterior probabilities, with lack of a number indicating maximum support. Silhouettes designed by ART

preferences, compared to other available models[28]. Accordingly, all our Bayesian analyses were performed using the compositionally site-heterogeneous CAT-GTR + G model. We also performed Maximum Likelihood (ML) analyses; however, CAT-GTR + G is not implemented in ML software. Accordingly, our ML analyses used LG + I + G4 (Matrix A and B) and LG + F + I + G4 (Matrix C), which emerged as the best fitting site-homogeneous models (according to ModelFinder[29]) for Matrices A, B and C, respectively, but only among the limited set of predefined empirical GTR models (see ref. [24] for a discussion) we tested in IQTree[30]. Matrix D proved too large to be analysed under CAT-GTR + G. Accordingly, this dataset was only analysed under ML in IQTree under the LG + F + I + G4 model (selected using ModelFinder). Given that for the ML analyses the overall best fitting model (CAT-GTR + G) could not be used (see above), and less fitting models had to be used instead, we suggest that when our ML and Bayesian results disagree, the latter should be preferred.

**Phylogenetic patterns in Chelicerata**. We found that CAT-GTR + G analyses of both Matrices A and B invariably support monophyletic Acari (Figs. 1b, c and 2). CAT-GTR + G analyses

of Matrix C did not converge, perhaps because Matrix C is derived from Matrix B but excludes the less saturated genes from Matrix A (Supplementary Table 2). The less saturated Matrix A supports the horseshoe crabs as the sister of monophyletic Arachnida (Figs. 1b and 2), whereas the more saturated Matrix B nested horseshoe crabs within the arachnids (Fig. 1c). Analyses performed under CAT-GTR + G can model site-specific compositional heterogeneity, but lineage-specific compositional heterogeneity can also negatively affect phylogenetic results[28,31]. We analysed Matrix A, which we deem more reliable based on the saturation-plot analyses (see above), under the best fitting CAT-GTR + G model after Dayhoff-6 recoding, which mitigates lineage-specific compositional heterogeneity and allows CAT-GTR + G to achieve greater modelling adequacy[28,31]. Analyses of the Dayhoff-6 recoded version of Matrix A destabilised some of the relationships within Arachnida (Fig. 1d) and caused support values for some nodes to reduce significantly, indicating that some relationships in Figs. 1b, 2 might represent compositional attractions. Nonetheless, our CAT-GTR + G analyses of the Dayhoff-6 version of Matrix A confirmed support for monophyletic Acari and did not reject a possible relationship between the horseshoe crabs and Arachnida. On the contrary, a placement of the horseshoe crabs within Arachnida is rejected by our

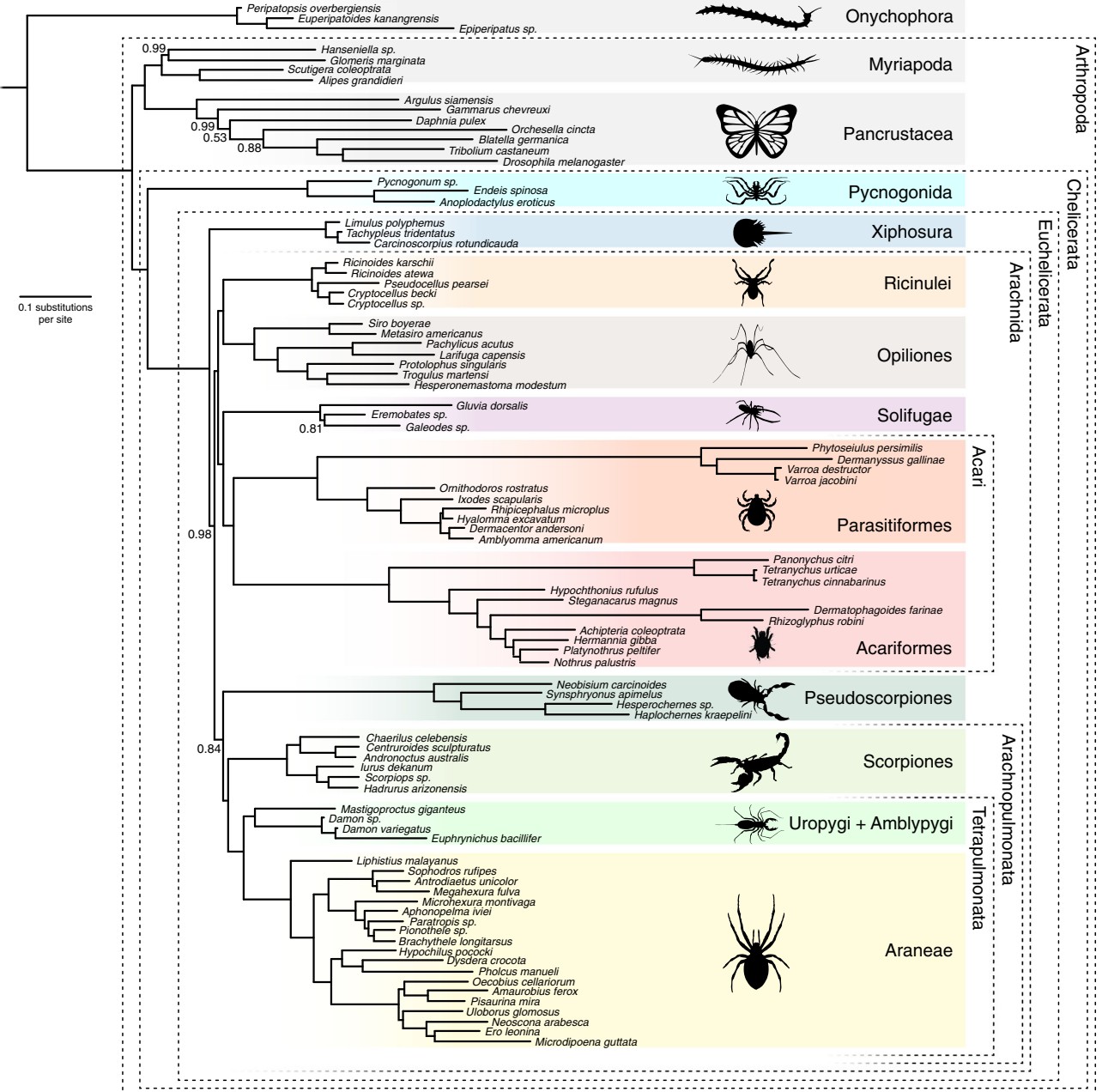

**Fig. 2** Phylogenetic tree derived from the CAT-GTR + G analyses of Matrix A after exclusion of six unstable taxa (*Calanus, Bothriurus, Liocheles, Pandinus, Superstitionia* and *Vietbocap*). Removal of unstable taxa did not affect topological relationships but increased support levels at key nodes and improved convergence statistics. Support values represent posterior probabilities. Convergence statistics: Burnin = 3000; Total Cycles = 10,000 subsampling frequency = 20. Maxdif = 0.28; Minimal effective size = 63. Silhouettes designed by ART

CAT-GTR + G analyses (with and without recoding) of Matrix A. The possibility that a placement of the horseshoe crabs within Arachnida[8,9] might be artefactual is further suggested by the observation that this topology is recovered from the more saturated Matrix B (Fig. 1c – irrespective of the model used), and from all four datasets under the less fitting LG + I + G4 (Matrices A and B) and LG + F + I + G4 (Matrix C and D) models (Supplementary Fig. 1 and Supplementary Fig. 2). In contrast, the consistent support for monophyletic Acari across all matrices, models and coding strategies, suggests that this node most likely represent phylogenetic signal, rejecting the rival Poecilophysidea hypothesis in which Solifugae (sun spiders) is the sister group of Acariformes alone[32–35], and a recently recovered clade in which Parasitiformes emerges as the sister group of Acariformes plus

Pseudoscorpiones[9]. However, Matrix A resolves Solifugae as the sister of a monophyletic Acari, still possibly indicating a close, even if not exclusive, relationship between the sun spiders and the Acariformes. In contrast, our CAT-GTR + G analyses invariably found Pseudoscorpiones in a close association with Arachnopulmonata (either as its sister group or as the sister group of Scorpiones). Pseudoscorpions were unstable in our ML analyses and emerged either as the sister group of Euchelicerata to the exception of Acari in Matrix A, B, and D (Supplementary Figs. 1A, 1B and 2B, respectively) or as a poorly supported sister group of Acari in Matrix C (Supplementary Fig. 1C). We are unaware of potential morphological synapomorphies for Acari and Pseudoscorpiones to the exclusion of Solifugae (e.g. the free finger of the chelicera articulating ventrally is shared by all three

groups[11]). In contrast, a potential morphological synapomorphy for Acari and Solifugae is a tight grouping of the epistomo-labral plate and discrete lateral lips around the anteriorly-positioned mouth, shared by this group to the exclusion of pseudoscorpions[36], and this grouping has been recovered in some morphological cladistic analyses (e.g. extant taxa cladogram of ref. [37]).

**Is Acari monophyletic?** Extensive lists of differences between acariforms and parasitiforms have been enumerated, and these entail numerous organ systems[12]. However, monophyletic Acari is still supported by more morphological apomorphies than any alternative set of relationships that have been suggested for both Acariformes and Parasitiformes. In addition, apomorphies supporting independent sets of sister group relationships for these taxa conflict with each other[11]. Morphologically, support for Acari has emphasized the gnathosomal feeding apparatus, a morphological unit at the front of the body composed of the chelicerae, the epistome, and the fused coxae of the pedipalps[38]. A gnathosoma has been variably regarded as shared by Acari and Ricinulei[39], but that of Acari exhibits a closer association of the chelicerae and palps and greater mobility of the complex as a whole[40]. Several other putative autapomorphies of Acari[39,41] are homoplastic with other arachnid orders (e.g. ingestion of solid food, aflagellate sperm, stalked spermatophores, an ovipositor), are reversals (e.g. absence of a pygidium), or have not been conclusively shown to be absent in Ricinulei (e.g. a hexapod prelarva). Some recent phylogenetic analyses of Chelicerata based on morphological characters have recovered monophyly of Acari when extant taxa are considered in isolation[40], or when fossils are included[42]. Alternatively, inclusion of fossils can result in Ricinulei shifting to an ingroup position[40], or either Solifugae or Ricinulei moving within a paraphyletic Acari[37]. Our analyses, including 10 species of Parasitiformes and 11 of Acariformes, converge on a sister-group relationship between these lineages, providing support for the monophyly of Acari based on genomic-scale datasets. Monophyly of Acari is the only current hypothesis for the relationships of the Parasitiformes and Acariformes that has consilient[2,16] support by genomic and morphological data. Within Acari, Parasitiformes comprises four major lineages, Mesostigmata, Ixodida, Holothyrida, and Opilioacarida, whereas Acariformes is composed by Trombidiformes and Sarcopti-formes. All analyses show a relationship between Phytoseioidea and Dermanyssoidea within Mesostigmata, and between Argasi-dae and Ixodidae within Ixodida. Relationships within Sarcopti-formes, on the other hand, could not be resolved with confidence as some alternative positions for an internal branch result in different definitions of Oribatida, either containing or excluding the Astigmata (Supplementary Fig. 3). We acknowledge that some key groups of Acari are not yet included in this or other transcriptomic analyses, such as Opilioacarida and Holothryida.

**Phylogenetic relationships within Arachnida.** All chelicerate taxa traditionally classified as orders are consistently retrieved as monophyletic across our analyses. Within Arachnida, all our analyses recover a monophyletic Acari. The sister group of Acari, however, could not be clearly identified. The CAT-GTR + G analyses of Matrix A found Solifugae in this position (but with variable support levels; PP = 0.77 in Fig. 1b and PP = 1 in Fig. 2). In addition, despite the existence of some morphological support for a clade of Acari and Solifugae[37], this clade was not recovered in the Dayhoff-6 recoded analysis of Matrix A. This suggests that grouping of Acari and Solifugae in our trees might represent a compositional artifact. In addition to that, the ML analyses of Matrix C suggested a close alliance between Acari and

Pseudoscorpiones, rather than Solifugae, although support for this node is very low. All our analyses recover a monophyletic Tetrapulmonata (Araneae sister to Uropygi and Amblypygi) in alliance with Scorpiones (Arachnopulmonata) or to a clade composed by Scorpiones + Pseudoscorpiones. Therefore, our results mostly support a single origin of book lungs, a hypothesis underpinned by detailed structural similarity between scorpion and tetrapulmonate book lungs[43], a shared ancestral whole-genome duplication[44], and also recovered in previous phyloge-nomic analyses[8,9]. In most instances in which Arachnopulmonata is retrieved, Pseudoscorpiones is found as its sister group. Together, these results suggest a close relationship between pseudoscorpions and arachnopulmonates (Fig. 2). Considerations of model fit favour a close relationship between Pseudoscorpiones and Arachnopulmonata. All CAT-GTR + G analyses of Matrices A and B, including the analyses of the Dayhoff-6 recoded version of Matrix A (Fig. 1b–d), support a sister group relationship between Opiliones and Ricinulei, in most instances with maximum support. This clade has not been supported by any morphological analysis, although early taxonomists sometimes classified Ricinulei as opilionids. The closest modern hypothesis is one that unites these two lineages with Acari[45]. A clade composed of Opiliones as sister group to Ricinulei and Solifugae was recovered in a dataset of the 500 slowest evolving genes in ref. [8], noting that these taxa share a fully segmented opisthosoma and tracheae. In most cases our ML analyses likewise support a relationship between Ricinulei and Solifugae, although always with low support (Supplementary Fig. 1A, B). Consideration of model fit (see above) leads us to suggest that Ricinulei plus Opiliones might be more likely to be correct. Intraordinal rela-tionships within the larger arachnid groups, such as in Opiliones or Araneae, are almost equivalent between different matrices and analyses.

**Summary.** We found support for clades that clarified key con-troversies in chelicerate phylogeny. Foremost among these is the alliance between mites and ticks, resulting in a grouping of ara-chnids with even more species than spiders. More broadly, our results suggest that the success of the arachnid order was most likely based on a single terrestrialisation event that happened after the last common ancestor of the horseshoe crabs diverged from the last common ancestor of Arachnida. Our analyses have also proposed some novel clades that need corroboration (such as a putative sister group relationship between Opiliones and Rici-nulei, and an alliance between Pseudoscorpiones and Arachno-pulmonata). We caution, however, that such lineages as Palpigradi and Opiliocarida remain unsampled. While further taxonomic sampling is needed to fully clarify the evolutionary history of chelicerates, consilience of genomic and morphological results marks a significant advance in our understanding of chelicerate evolution.

## Methods
**Data acquisition and transcriptome assembly**. The molecular matrices were composed of protein-coding genes from 95 species, mostly from Illumina tran-scriptomes, and largely retrieved from public repositories (Supplementary Table 1). We generated four new transcriptomes: the sea spider *Pycnogonum* sp., the soli-fugid *Galeodes* sp., the pseudoscorpion *Neobisium carcinoides*, and the amblypygid *Damon* sp. We have complied with all relevant ethical regulations for animal testing while collecting and processing these animals. For these four species, total RNA extractions were performed from single specimens using TRIzol® Reagent (ThermoFisher scientific) following the manufacturer's protocol. Transcriptome-wide cDNA libraries were generated and sequenced using Illumina HiseqII in Edinburgh to an estimated coverage of ~100×, using 100-bp paired end reads. These newly generated transcriptomes and those from the raw sequences down-loaded from the public repositories were assembled using Trinity version 2.0.3[46] under default parameters and using Trimmomatic (default parameters, as part of the Trinity package) for quality control. The new transcriptomes were deposited in

NCBI as Sequence Read Archive (National Center for Biotechnology Information) under the accession numbers PRJNA438779 (see Supplementary Table 1). All assemblies were processed by predicting the putative proteins from the assembly results, employing a previous filter of reduction of redundant isoforms using CD-HIT-EST with a 95% similarity cutoff[47]. These filtered results were processed in TransDecoder[48] in order to identify candidate open reading frames within the transcripts and translate them into proteins.

**Orthology assignment and matrix assembly.** We generated three alternative molecular datasets based on transcriptomic data for 95 species. The first super-matrix, Matrix A, contained 233 genes and represents an extension of a dataset we previously assembled[2,16,17]. Genes in this dataset were selected based on being largely single copy and presenting a slow rate of evolution. The taxonomic sample spanned 80 chelicerates, 21 of them representing Acari, and 15 outgroups. New chelicerate sequences matching those in the considered dataset were acquired through BLAST[49] searches over the set of proteins translated from the transcriptomic sequences. We used sequences from *Daphnia pulex as* reference seeds for our orthology mining process, as this species has full coverage for the 233 considered orthologs in this dataset. MoSuMa[3,50], a custom Perl pipeline (available at github.com/jairly/MoSuMa_tools/) can be used to automatically expand pre-existing phylogenomic datasets. To do so, MoSuMa performs a variety of steps the first of which is a BLAST search from which it selects a subset of putative orthologs. For this step, the best hit is chosen together with all the sequences with an e-value within three orders of magnitude of the top hit. These suboptimal sequences are retained to provide possible alternative orthologs. We set the maximum (worst) acceptable e-value to be $e = 10^{-10}$, with hits $e > 10^{-10}$ being automatically rejected. For each considered protein family, MoSuMa aligns all putative selected orthologs using MUSCLE[51] (applying default parameters), against a pre-existing gene alignment. For our dataset, this generated 233 gene-alignments. Gene trees were then inferred in IQTree[30], applying the model of best fit as assigned by ModelFinder[52]. For nearly all gene trees, the model LG + I + G4 was found to be best fit. The 233 gene trees were assessed using a custom Perl script (/github.com/jairly/MoSuMa_tools/blob/master/treecleaner.pl), following the criteria defined in Lozano-Fernandez et al.[3], to exclude possible paralogs and putative orthologues displaying anomalously long branches. Each gene tree was then visually inspected to evaluate whether MoSuMa might have inadvertently failed to remove putative paralogs[3]. The final gene alignments, cleaned of long-branched sequences and putative paralogs, were concatenated using SequenceMatrix v100[53]. Finally, we removed ambiguously aligned positions using GBlocks v0.91b[24]—parameters $b2 = 70\%$, $b3 = 10$, $b4 = 5$, $b5 =$ half. This process generated a final supermatrix of 45,939 amino acid positions across 95 taxa, with a 78.1% level of completeness. We call this dataset "Matrix A".

A second dataset, Matrix B, was compiled de novo. To assemble this dataset, we first used full-orthology assessment (across a set of 23 taxa—marked in bold in Supplementary Table 1) as implemented in the OMA stand-alone software[21]. To this initial 23-taxon dataset, we added orthologues for a further 72 species using MoSuMa (see below for details). In contrast to Matrix A (see refs. [2,16,17]), Matrix B was compiled without attempting to filter out genes based on their expected phylogenetic utility.

To limit computational time, the OMA analysis was limited in the number of transcriptomes initially used to generate the orthologous groups. For this initial phase only 23 high-quality transcriptomes (marked in bold in Supplementary Table 1) were considered. More precisely we used: one transcriptome for each outgroup clade, one for each chelicerate order represented in the final (95 taxa) dataset by less than five taxa, two transcriptomes for each chelicerate order represented by five to 10 taxa in the final dataset, and three transcriptomes for each order represented by more than 10 taxa in the final dataset. Based on this initial set of 23 transcriptomes OMA identified 25,473 orthologous groups. However, many of these orthogroups had low occupancy across taxa, and we retained only those present in more than 35% of the taxa to avoid creating a dataset dominated by missing data. This higher occupancy dataset included 3982 orthogroups. For the 3982 retained orthogroups, we identified orthologs in the other 72 transcriptomes considered for our study using MoSuMa. For all the steps in MoSuMa (except the Gblocks analysis) we used the same parameters used to generate Matrix A. The final gene alignments were concatenated using SequenceMatrix v100[53] generating a matrix of 2,006,612 amino acids. However, to reduce noise from potentially misaligned positions in Matrix B (also given that for this much larger dataset gene trees could not be inspected) we trimmed this matrix using more stringent settings in Gblocks v0.91b[24]—parameters $b2 = 50\%$, $b3 = 20$, $b4 = 2$, $b5 =$ none; which generated a final matrix of 34,839 amino acid positions (completeness = 86.4%). While the initial dataset consisted of 3982 loci, the stringent trimming approach implemented on this dataset most likely resulted in a reduction in the total number of represented loci.

To evaluate whether the use of MoSuMa might have affected the assembly of Matrix B, we compared differences in orthology assignments between MoSuMa and OMA for the genes over which Matrices A and B overlap (204 genes, see Supplementary Table 2). We selected the 10 orthogroups shared between these matrices with the best and worst gene coverage in OMA across 23 taxa. For the genes with best coverage, we found that for the 230 cases analysed (inspecting the 23 taxa used in the OMA analyses that are also in Matrix A), OMA and MoSuMa

retrieved identical orthologous genes for each species in 82% of cases. In 17% of the cases, only one of the methods retrieved an ortholog. Only in 0.8% of cases (two instances out of 230) did OMA and MoSuMa disagree on the protein assigned as the ortholog. For the loci of lowest coverage similar results were found. In 70% of the cases both methods retrieved the same ortholog (or absence of it), and in 30% of the cases, only one of the two methods retrieved an ortholog. When using the 10 orthogroups of lowest coverage, not a single case was identified where the two orthology assignment methods disagree (see Supplementary Table 3).

Matrix C was generated removing the genes present in Matrix A from Matrix B (see Supplementary Table 2), so that Matrices A and C constitute two independent datasets, derived from non-overlapping gene sets. Matrix C is composed by 3778 OMA-defined ortholog groups, trimmed using the same parameters applied in Matrix B, and yielding a final dataset of 30,552 amino acid positions (completeness = 86%). To test whether the stringent trimming we imposed on matrix B and C affected our results we generated a larger dataset starting from the set of loci used to derive Matrix B but applying a softer trimming strategy. We used this dataset to test also the effect of the trimming software used on phylogenetic inference. To achieve both goals we used TrimAl[25] (instead of Gblocks[24]) with the option -gt = 0.9. This retained sites with 10% or less missing data and results in an alignment of 127,114 amino acids positions (3.6 times larger than Matrix B). This dataset, Matrix D, was analysed using ML and recovered a topology that was in agreement with those obtained from our other ML analyses, except that the five species with the worst quality transcriptomes (two onychophorans, two sea spiders, and a whip spider—see Supplementary Table 1) formed a clearly artificial group nested inside Solifugae (Supplementary Fig. 2A). Matrix D was thus reanalysed after having removed these taxa, and we only discuss in details the results of this taxonomically reduced version of Matrix D (Supplementary Fig. 2B).

All our matrices are available at https://bitbucket.org/bzxdp/lozano_fernandez_2019.

**Substitution saturation analysis.** APE[54] was used to calculate taxon-to-taxon (i.e. patristic) distances for the ML trees derived from Matrices A, B, and C. These distances were plotted against uncorrected observed distances generated for the same matrices in PAUP4.0a[55]. When deriving saturation plots the expectation is that uncorrected genetic distances will more strongly underestimate true evolutionary distances as saturation increases[23], because these distances do not account for multiple substitutions. Accordingly, uncorrected observed distances will correlate more poorly with patristic distances derived from a ML tree derived using substitution models (in our case LG + I + G4 and LG + F + I + G4) that allow the estimation of multiple substitutions per site. In a saturation plot, the lower the $R^2$ and the slope, the greater the saturation of the considered dataset.

**Phylogenetic analyses.** We performed phylogenetic analyses using both Maximum Likelihood and Bayesian Inference. All maximum likelihood analyses were completed in IQTree[30] under the LG + I + G4 model. All Bayesian analyses were completed in PhyloBayes MPI v1.5a[56] under the CAT-GTR + G model. Cross-validation was carried out to assess model fit on our least saturated dataset (Matrix A), as described in the PhyloBayes manual. For the model fit analysis[26] we compared GTR + G against CAT-GTR + G. CAT-GTR + G emerged as outperforming GTR + G, with a score of 21.1 ± 8.39 (in favour of CAT-GTR + G), and across all ten cross-validation replicates. For the IQTree analyses we used LG + I + G4 (Matrix A and B) and LG + F + I + G4 (Matrix C) selected as best fit using ModelFinder[52]. As LG is an empirical GTR matrix that invariably fits the data worse than a dataset-specific GTR + G, we can safely conclude that CAT-GTR + G represents the overall best fit model, and that therefore the LG-based ML trees should be considered less reliable than the CAT-GTR + G-based Bayesian trees when phylogenetic relationships between these trees differ. Matrices A, B, and C were analysed using ML and Bayesian analysis at the amino acid level, but in Matrix C the CAT-GTR + G Bayesian analysis did not converge, possibly due to a deterioration of the signal-to-noise ratio caused by the fact that the less saturated genes in Matrix A (Fig. 1a) are not in Matrix C.

To assess the potential impact of lineage-specific compositional heterogeneity on our results we also analysed Matrix A after Dayhoff-6 recoding using the CAT-GTR + G model of amino acid substitution[27]. Dayhoff-6 recodes the 20 different amino acids into six groups on the basis of their chemical and physical properties. This approach excludes (frequent) replacements between similar amino acids and reduces the effects of saturation and compositional bias[27].

As customary, for all Phylo Bayes analyses two independent runs were completed. Convergence was tested using the *bpcomp* and *tracecomp* programs in PhyloBayes (statistics reported in corresponding Supplementary Figures). Support in our Bayesian trees represents Posterior Probabilities. Support values in the ML trees are bootstrap proportions. Bootstrap analyses in IQTree used 1000 replicates and the ultrafast inference method[57].

**Reporting summary.** Further information on research design is available in the Nature Research Reporting Summary linked to this article.

## Data availability

All scripts, individual gene alignments, amino acid matrices, and phylogenetic trees have been deposited as a Bitbucket repository and can be accessed at https://bitbucket.org/bzxdp/lozano_fernandez_2019. The transcriptomes generated as part of our study are available at NCBI Sequence Read Archive – BioProject PRJNA438779. Individual SRA numbers for the raw read data of each species are listed in Supplementary Table 1. The source data underlying Fig. 1a are provided as a Source Data file.

## Code availability

All custom codes used in the generation and processing of data are part of the pipeline for the compilation of molecular supermatrices, MoSuMa[8,9], a custom Perl pipeline available at www.github.com/jairly/MoSuMa_tools/.

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

## Acknowledgements

We thank Jason Dunlop for kindly providing some of the sequenced samples, and David Walter for comments on the manuscript. We also thank Rosa Fernández, Prashant Sharma and Gonzalo Giribet for sharing sequence data information. This research was funded by a Marie Skłodowska-Curie Fellowship (655814 to J.L-F.), a NERC BETR grant (NE/P013678/1 to DP and JV), and the European Union's Horizon 2020 research and innovation programme under the Marie Skłodowska-Curie grant agreement 764840 to DP and MG (ITN IGNITE – http://www.itn-ignite.eu/). A.R.T. was supported by a University of Bristol (STAR) PhD studentship, M.G. is supported by a Marie Curie (ITN – IGNITE) PhD studentship, R.C. was supported by a Science Foundation Ireland grant to DP (11/RFP/EOB/3106).

## Author contributions

Experiments were designed by J.L-F., A.R.T., D.P., G.D.E., and J.V. The main text was authored by J.L-F., A.R.T., D.P., and G.D.E., with further suggestions from all authors. Sequencing laboratory work was carried out by J.L-F. Matrix compilation and computational analyses were carried out by A.R.T., J.L-F., and M.G. The new generated transcriptomes were assembled by R.C. Figures were designed by A.R.T.

## Additional information

**Competing interests:** The authors declare no competing interests.

