## [Peer Review File · Nature Communications]

Reviewers' Comments:

Reviewer #1:

Remarks to the Author:

This is a potentially significant study offering novel insights into arachnid phylogeny. Mites and ticks together make up the majority of the arachnid species, but their relationships to other orders have long been problematic. As the authors note, there has been a trend in recent years to treat the two main branches of the old order Acari as independent orders (Acariformes and Parasitiformes) and several lines of evidence have suggested that Acari may not be a monophyletic group. The present study is one of the few molecular phylogenies to offer explicit evidence that Acari in its traditional sense could indeed be monophyletic, and in this sense it merits inclusion in a high-profile journal such as Nature Communications.

However, I would like to raise some issues - mostly relating to taxon sampling - before recommending publication.

TITLE

I'm not sure if the phrase 'mites and ticks' is the best choice here since ticks are only one part of the much wider parasitiform clade. I realise that 'mites and ticks' sounds better to a non-specialist reader, but technically the authors should be saying "...supports the monophyly of Acari...". Perhaps the Editors have a preference here?

Also, I'm not sure in what sense the authors are reconciling molecules and morphology. The morphological data presented here is rather superficial. The gnathosoma is the only synapomorphy offered here in support of (Acariformes + Parasitiformes), other traditional apomorphies like the hexapodal larva are not mentioned, and the (numerous) anatomical differences between the two mite clades listed by, e.g. Dunlop & Alberti, are not critically re-examined or otherwise considered in any detail. Essentially, this is a molecular phylogeny only and it would perhaps be fairer to present it as such.

INTRODUCTION

I think the authors need to be a little careful when talking about orders. I realise this is a semantic and to some extent artificial construct. The authors mention 12 orders in the Introduction which implies that Acariformes and Parasitiformes are two of these orders. Table S1 also lists Acariformes and Parasitiformes specifically as orders. However the Introduction also says (p. 2) that Acari would be "...the most speciose arachnid order." The authors thus need to decide whether, in the light of their results, they recognise a single order Acari or a clade Acari comprising two orders: Acariformes / Parasitiformes. Personally I would favour the latter option given that there are still some fairly fundamental differences between these major mite groups.

RESULTS AND DISCUSSION

As I think the authors may recognise themselves, the most significant weakness of this study is its limited taxon sampling. While this in itself should not preclude the publication of novel and interesting data, I think the authors might want to be a little less bullish and confident in their results. It's the difference between saying, for example, "Acari is monophyletic" and "Acari is recovered as monophyletic"

In detail, the authors have sampled about 20 mites which is far less than, say, the ca. 90 ingroup mites of the Pepato & Klimov (2015) study or the 142 ingroup mites sampled by Dabert et al. (2016); both of whom did NOT recover a monophyletic Acari!

I'm not disputing the present author's results, but I think they are being disingenous to reject previous (molecular) studies which did not recover a monophyletic Acari without explaining clearly why their data, from a much more limited sample of taxa, is likley to be giving a more reliable result. Is the fact that they are using genomic-scale datasets the key difference here? Is there precedent for these methods being fundamentally better for phylogenetic reconstruction?

I also note that two key mite groups (Opilioacarida and Holothyrida) were not sampled at all. Opilioacarids are important in that they are widely seen as 'primitive' mites retaining several plesiomorphic characters for arachnids. Holothyrids are interesting as the putative sister-group of the ticks. I realise that material of these rare mites may not have been available, but I would have expected the authors to at least note that these lineages could not be sampled and thus to be a little more circumspect in presenting their results.

Another key non-mite group which is missing here are the palpigrades. Often promoted as 'primitive' arachnids, in some phylogenies (e.g. van der Hammen 1989) they were interpreted as closely related to at least the acariform mites and are thus of some relevance to the hypotheses being tested here. The lack of data for this taxon should also be conceded/noted in the Results and Discussion and the implications of this considered.

I would also like to stress that the mites that are sampled are taxonomically very localised (three tetranychoids: two in the same genus *Tetranychus*), two species of the highly derived parasite *Varroa*, etc. In this context I don't think the manuscript really contributes much to the study of relationships within the mites (as per Suppl. Fig. 2). The phytoseiid/dermanyoid and the argasid/ixodid pairings are non-controversial and not actually very interesting here.

The monophyly of Trombidiformes and Sarcoptiformes respectively is hardly being tested here with the eleven acariform terminal taxa used. Most of the major trombidiform lineages (*Spaherolichida*, *Labidostomatides*, *Eupodides*, *Anystina*, *Parasitengona*, *Heterostigmata*) are missing. In this context I think that calling this "...taxonomically well-sampled datasets..." in the Abstract (p. 1) is somewhat misleading.

Given that the main conclusion is that Acari is monophyletic, I would have expected a little more discussion about what the sister group of Acari is; particularly in the section 'Phylogenetic relationships within Arachnida'. Implicitly the authors recover (Acari+Solifugae), and might want to discuss whether this has any morphological support. As they note, several characters already support Acariformes+Solifugae, but do ALL mites and camel spiders have anything in common?

Off the top of my head, I can think of the epistomo-labral plate and the fact that the free finger of the chelicera articulates ventrally against the fixed finger; although these characters may also be present in pseudoscorpions as well, e.g.

Dunlop, J. A. 2000. The epistomo-labral plate and lateral lips in solifuges, pseudoscorpions and mites. In Gajdos, P. & Pekár, S. (eds). Proceedings of the 18th European Colloquium of Arachnology, Stará Lesná, 1999. *Ekológia* (Bratislavia), 19 (supplement 3/2000): 67–78

The authors do briefly discuss Opiliones + Ricinulei, although I feel that this is an unconvincing clade with little or no morphological support. The only thing in common I can think of is an elongate second

pair of legs, but this is not present in the basal cyphophthalmid harvestmen and is probably an Ingroup character for Opiliones. That said, prior to the Hansen & Soerensen (1904) monograph "On two orders of Arachnida", it might be worth noting that ricinulids were traditionally regarded as unusual harvestmen.

In summary, I would recommend that the authors (a) more clearly acknowledge the gaps in their taxon sampling, (b) make clearer why their results from a more limited dataset are likely to be more reliable than those of previous studies and (c) discuss in more detail - preferably with some morphological characters - the relationships between the arachnid orders recovered in the present work; especially the putative sister group of their monophyletic Acari.

MINOR CORRECTIONS

p. 1: "hemophagic" or (p. 2) "haemophages"? Please check British versus US spelling and be consistent.

p. 2: According to version 3 of "A Manual of Acarology" (Krantz & Walter, 2009) there are about 55,000 described mite species.

p. 7: Supplementary Information better "...with focus on the..."

p. 8: don't capitalise titles of papers, i.e. "A Review of the fossil record of spiders (Araneae) with special reference to Africa. and description of a new specimen..."

p. 8/9: Giribet's 2018 paper has now been formally published with volume and page numbers of "237, 7-13".

p. 9: Should it be "Sperling, E. A." [not E. a.]?

p. 9: should it be "Lake, J. A." [not J. a.]?

p. 10: "An asymptotic equivalence of choice model by cross-validation and Akaike's criterion"

p. 10: "...relationship with Pseudoscorpiones (Arachnida)."

p. 11: I think if you want to cite Weygoldt and Paulus's cladistic results you should really refer to the second part of their paper, i.e.

Weygoldt, P. & Paulus, H.F. Untersuchungen zur Morphologie, Taxonomie und Phylogenie der Chelicerata: II. Cladogramme und die Entfaltung der Chelicerata. Z. Zool. Syst. Evol. 17, 177–200 (1979).

Please note that when this paper was published the journal had a German title.

p. 11: "Cross-validators choice and assessment of statistical predictions."

MINOR CORRECTIONS: SUPPLEMENTARY DATA

p. 4: Italicise the journal name "Nat. Protoc."

p. 5: I think the more usual Journal abbreviation is "Phil. Trans. R. Soc. B"

p. 5: "Improved modeling of compositional heterogeneity supports sponges as sister to all other animals."

TABLE S1

Is there any particular reason why two Illumina sources (for *P. citri* & *R. robini*) are italicised.

Tetranychus cinnabarinus [not *Cinnabarinus*]

Phytoseiulus [not *Phytoseiulus*]

I do not mind my identity as a reviewer being revealed
Jason A. Dunlop, Berlin

Reviewer #2:

Remarks to the Author:

Review Lozano-Fernandez et al.

In their manuscript entitled 'Genome-scale chelicerate datasets support the monophyly of mites and ticks, reconciling molecules and morphology' the authors describe a phylogenomic approach to address the evolutionary relationships among chelicerate orders. In contrast to previous studies with the same – or at least a similar – aim, Lozano-Fernandez et al. have compiled a data set of unprecedented comprehensiveness encompassing all but two of the known chelicerate orders as well as more slowly evolving representatives of the Acari. They separate their data into two different data matrices, one – data set A – encompassing 234 loci and 45,939 alignment columns, and the other – data set B – encompassing 3,982 loci and 34,839 alignment columns. On this basis, they then tackle the problem to resolve the backbone of the chelicerate phylogeny. Specifically, two main questions are addressed: (i) Do mites and ticks can be grouped into monophyletic Acari, and (ii) what is the phylogenetic placement of the horseshoe crabs. From their analysis, three results emerge. First, a well selected data set is necessary from which an interpretable phylogenetic signal can be extracted. Second, what is left on phylogenetic signal even in the hand-selected data requires Phylobayes in combination with the CAT-GTR-G model to be correctly interpreted. And third, a tree is presented that reflects what morphologists have claimed for quite some time: Mites and ticks are monophyletic, and horseshoe crabs form the sister of the terrestrial chelicerates suggesting only a single transition from aquatic to terrestrial life style.

In a strict sense, the manuscript provides a little less novelty than it seems on the first sight. Basically, the two methodological issues have been already extensively discussed in the literature, and this study adds no further points that could help to further shed light on this point. This leaves the insights into the chelicerate phylogeny as the main driving point. And indeed, the authors claim that they could clarify some of the controversies that exist in the literature with respect to the intraordinal relationships within the chelicerates. This reasoning is one of the biggest concerns I have with this manuscript. Reading the manuscript a bit naively, one may get the impression that the claim is based on eventually – and after many trials – observing a phylogenomic tree that is congruent with what

morphological data told us for quite some time. Obviously, this would be poor reasoning because it would make the morphological evidences the gold standard. I, therefore, would wish for a more objective approach of the problem where the results of the phylogenomic analyses are presented in a way such that they can stand for themselves. Only then the comparison to the morphological data should be performed. If this can be achieved, the phylogenomics studies can then indeed provide another line of evidence that help in resolving the early stages of chelicerate evolution.

Major issues

1. The authors present two data sets of which they consider one, data set A, superior to the other. I think having two non-overlapping data sets telling a consistent evolutionary story would be a relevant asset to this study. It is therefore sad that data set B earns very little attention and no attempt is made to improve it over what was achieved with the standard procedures. Specifically I wonder
 - a. What is the overlap between data sets A and B. Somehow I can imagine that data set A should be a subset of data set B. If so, this overlap should be removed to have two independent views on the same process. If they do not overlap with respect to the represented loci, I would be highly interested why this should not be the case.
 - b. The authors mention the higher level of saturation in data set B compared to A and explain with this the poor performance in the tree reconstruction. Looking at figure 1A, I wonder why no attempt was done to decrease the level of saturation in this data set by removing those genes that are particularly effected. The authors even write in the manuscript that no attempt was done to do so. I am a bit puzzled that – given the severity of the reconstruction problem – not all attempts are done to maximize the information content in the data. As already stated above, it would be highly desirable to have a consistency criterion at hand that can be used to generate confidence in the phylogenomic reconstruction independent from morphological evidences.
 - c. I am wondering about the nature of data set B. The authors state that they considered almost 4,000 loci (orthologous groups) in this data matrix. Still, the alignment is not even 35,000 amino acids long. Thus, the average contribution per locus is less than 10 amino acids. I think this clearly shows that there is ample room for the improvement of this data set. Note, the data set A comprises only a little more than 200 genes but is about 10,000 amino acids longer.
 - d. The 'orthology' assignment procedure used in this study is an interesting approach that requires a massive amount of human interaction. I am wondering why not a published approach for an automated extension of orthologous groups with further sequences from transcript data was applied. Moreover, the authors state in the supplement that they visually inspected each tree and each candidate ortholog in order to make the decision to keep a candidate, or not. Somehow I cannot believe that this procedure was done for all 3,982 orthologous groups that are considered in data matrix B. Lastly, how can one rule out that this visual inspection and sequence selection does not introduce an unwanted bias by favoring those orthologs that place a taxon at a certain position?
 - e. The data matrix B was compiled using an integrated approach that combines OMA and the authors' own custom approach for ortholog search. This combination was apparently necessary to cope with the computational burden. Two questions arise: First, I trust that in some cases there should be ortholog assignments available both from OMA and from the "BLAST-based approach". It would be interesting to know in how many cases they differ. Second, the OMA approach requires a subselection of taxa from which the orthologous groups were collected. How were these taxa selected?
2. In their phylogenomic analysis, the authors find at least one grouping that is consistently observed across all trees. Most prominently, this is the alliance of Opiliones and Ricinulei in a sister clade. Somehow, I have the impression that this true novelty earns surprisingly little attention in the discussion. One even gets the impression that the authors are not confident that this grouping is not an artefact, which probably is reflected in the sentence that these groupings require 'corroboration'. If anything, I would think that this study – if the authors consider it comprehensive – should provide the basis for coming up with new hypotheses.

In addition I see a number of minor yet essential issues.

1. The authors mention that they, in the ML framework, used LG+G+I as a model for sequence evolution. Could it be true that the F parameter for estimating the amino acid frequencies from the data was not suggested when the model with the best fit was searched with ModelFinder?
2. The authors extracted at least part of their data from transcriptome reconstructions. Although I know that the terminology is not strict in this context, I think it is worthwhile making a distinction between gene sets (called "proteomes" now and then) inferred from whole genome sequences, and those sets that have been inferred from transcriptomes. Reason is, we will – by necessity – miss always very lowly expressed genes and those that are not transcribed at all in our transcriptomes. Unfortunately, it is one of the inherent assumption of the ortholog search tools, such as OMA, that they have the full gene set at hand for the inference.
3. On page 4, the authors state that their cross validation test revealed that the LG+G+I was not the overall best fitting model and refer to something that was written further up in the text. I find further up no further mentioning of the cross validation in the context of the LG model. This should be clarified. Furthermore, it might be worthwhile to be a bit more precise in the wording as also readers not professionals in phylogeny reconstruction in a likelihood framework should be able to follow the argumentation.
4. In the methods, the authors explain why they used *D. pulex* sequences as a 'bites'.
 - a. Probably most of the readers will not be familiar with the custom approach used here to predict orthologs in transcript data. Hence, they will not be familiar with the particular terminology, such as 'bites'.
 - b. *D. pulex* was used because it has "phylogenetic proximity" to the chelicerates. I would appreciate if this could be elaborated a bit further. *D. melanogaster*, just as any other insect, should have the same level of "phylogenetic proximity", is this correct? But probably this is not what the authors have in mind?
5. Figure 1A – the caption is a bit sparse. What do the individual dots represent? Not every reader is familiar with such plots.
6. Figure 1B-D. There a quite a number of branches without support values. If this is maximal support, then this should be mentioned in the legend.
7. Figure 2 – The legend is very specific with respect to details of the Bayesian analysis. I trust that quite a bit of this can go into the methods section (Burn-in, total cycles, subsampling frequency).
8. Although the analysis focusses on the interorder relationships, I wonder to what extent the topologies within the individual orders agree (or disagree) between the different trees. Somehow one would expect that these are a bit less problematic to resolve, and hence we would expect the trees to be largely congruent. Is this indeed the case? If not, what would be a possible explanation for any incongruence?

Reviewer #3:

Remarks to the Author:

In the present manuscript Lozano-Fernandez and co-authors study the relationships of some (most) chelicerate lineages using primarily available transcriptomic datasets. Data for additional 4 taxa is generated as part of this study. Although the taxon sample is rather wide and the first part of the title is more general, the authors mainly focus on one particular results, which is the monophyly of acari (a lineage that includes both, mites and ticks). That is indeed an interesting finding given the general lack of support of monophyletic acari based on molecular data although there are numerous morphological similarities that have traditionally supported this group (as mentioned by the authors).

Few other results are also mentioned briefly (e.g., the sister group relationship between Opiliones and Ricinulei) these are not discussed much.

Although the finding of monophyletic Acari based on molecular data is an intriguing result I have several concerns regarding the ms. These are listed here in no particular order.

I guess my main concern is the supposed high support for acari and the overall novelty and significance of this finding. This clades is indeed present in the results but support is no that high in several cases (e.g., Fig 1B, D). When support values are not shown I assume that means full support (100%) but this is not stated anywhere explicitly but 0.68 is not high and 0.74 can be considered but I would not call it high. Furthermore, the authors state that they have checked Phylobayes runs for convergence and ESS and have made sure that these are within the acceptable limits (ESS>50 and maxdiff values <0.3 which is the bare minimum) but they do not provide the statistics that were found for their analyses. So if ESS were rather close to 50 and maxdiff to 0.3 I would strongly suggests that the authors make an effort to improve these. In any case, I think that ESS, maxdiff and other similar statistics should be provided (e.g. in a suplementry table, text etc.).

The authors also do not provide much discussion on what previous molecular studies have found regarding acari and how this compares with the results presented here. What is the broad impact of this finding? Why it is important in a boarder context? Nature Communications is a multidisciplinary journal with a extensive readership so putting the present findings in a broader context is very relevant here. What about previous findings on the internal structure of the two main linages within acari? This is basically missing in the discussion.

The use of the word diversification in the third subheading of the results is not appropriate. Relationships are discussed in this section but not diversification patterns. There are no diversification analyses either.

There is a mention of what morphological data has suggested in terms of the Opiliones and Ricinulei but what about molecular data? Previous analyses based on molecular data put Opiliones in the vicinity of Ricinulei so that is not that unexpected finding here. I think that some more discussion and references are needed here.

When the authors talk about the use of models in the ML analyses (last lines of page 4) they use the term "overall best fitting model". What "overall best fitting model means? You state that LG+G+I was the best-fit model selected with ModelFinder for the ML analyses some lines above yet it is not "overall best fitting" Further, you state "we suggest that when results from our LG+G+I and CAT-GTR+G analyses (with and without Dayhoff recoding) disagree, the latter should be considered more reliable." What would be the rationale for that? Selecting best fit models is based on a quantitative approach (you heave used ModelFinder). Here you suggest that when model selections finds a model that result a topology that does not fit a "preferred" case (as in the LG+G+I analyses) model that leads to results that comply with this "preferred" topology should be favored.

But what is the objective quantitative criterion that you suggest? I think that the statements made in this sentence need to be explained better and with more clarity. As it is, I do not find this very convincing.

On Figure 1 A It looks like the low R square for matrix B is driven by relatively low number of outliers. I wonder how results might look if these positions are excluded and then Matrix B is used. In such way the dataset would be still larger than the matrix A and would not be subjected to an apriory data selection process.

Finally, text labels (e.g., tip labels) on supplementary figures hard to read.

The supplementary text in places need closer revision of stile/grammar.

In conclusion, I think that the novelty and the broader significance of the purportedly novel finding of monophyletic Acari (based on molecular data) are not well presented and discussed. I also have some concerns regarding the level of support for Acari and the convergence/ESS for the Bayesian analyses. Thus I can not recommend publication of this ms in its current shape.

However, I think that the presented results are interesting and there is a potential, after a proper review, for this ms to fit the type of studies that Nature Communications aims at. Thus, my recommendation is rejection with invitation to resubmit a revised version.

Reviewers' comments:

Reviewer #1 (Remarks to the Author):

This is a potentially significant study offering novel insights into arachnid phylogeny. Mites and ticks together make up the majority of the arachnid species, but their relationships to other orders have long been problematic. As the authors note, there has been a trend in recent years to treat the two main branches of the old order Acari as independent orders (Acariformes and Parasitiformes) and several lines of evidence have suggested that Acari may not be a monophyletic group. The present study is one of the few molecular phylogenies to offer explicit evidence that Acari in its traditional sense could indeed be monophyletic, and in this sense it merits inclusion in a high-profile journal such as Nature Communications.

However, I would like to raise some issues - mostly relating to taxon sampling - before recommending publication.

TITLE

I'm not sure if the phrase 'mites and ticks' is the best choice here since ticks are only one part of the much wider parasitiform clade. I realise that 'mites and ticks' sounds better to a non-specialist reader, but technically the authors should be saying "...supports the monophyly of Acari...". Perhaps the Editors have a preference here?

We changed this to say "...supports the monophyly of Acari", as recommended.

Also, I'm not sure in what sense the authors are reconciling molecules and morphology. The morphological data presented here is rather superficial. The gnathosoma is the only synapomorphy offered here in support of (Acariformes + Parasitiformes), other traditional apomorphies like the hexapodal larva are not mentioned, and the (numerous) anatomical differences between the two mite clades listed by, e.g. Dunlop & Alberti, are not critically re-examined or otherwise considered in any detail. Essentially, this is a molecular phylogeny only and it would perhaps be fairer to present it as such.

We agree, and we recast this argument to clarify that our study is based on the study of molecular data and the consistency of our molecular results with some previous morphological arguments. We appropriately cited references to our previous works also based on the use of consistency as a mean to corroborate and reject hypotheses in a phylogenetic framework (Rota-Stabelli et al. 2011 Proc B; Campbell et al. 2011 PNAS). We have also de-emphasised the "reconciling" argument to reflect the fact that Acari is not emphatically the morphological solution. We have added details on many of the other proposed apomorphies of Acari but also indicated that they are generally homoplastic with other arachnid orders or have not been adequately shown to be lacking in some orders (we now cite the hexapod prelarva in this regard). Gerd Alberti's extensive list of (77!) differences between the two subclades of Acari is acknowledged and cited (even though differences are irrelevant in a cladistic framework where only shared similarities define clades), but together with the caveat that despite these differences, few characters support rival sister groups for the two subclades of Acari.

INTRODUCTION

I think the authors need to be a little careful when talking about orders. I realise this is a semantic and to some extent artificial construct. The authors mention 12 orders in the Introduction which implies that Acariformes and Parasitiformes are two of these orders. Table S1 also lists Acariformes and Parasitiformes specifically as orders. However the Introduction also says (p. 2) that Acari would be "...the most speciose arachnid order." The authors thus need to decide whether, in the light of their results, they recognise a single order Acari or a clade Acari comprising two orders: Acariformes / Parasitiformes. Personally I would favour the latter option given that there are still some fairly fundamental differences between these major mite groups.

We agree and we have tried to rephrase as necessary to replace order(s) with group/lineages/clades etc throughout the text.

RESULTS AND DISCUSSION

As I think the authors may recognise themselves, the most significant weakness of this study is its limited taxon sampling. While this in itself should not preclude the publication of novel and interesting data, I think the authors might want to be a little less bullish and confident in their results. It's the difference between saying, for example, "Acari is monophyletic" and "Acari is recovered as monophyletic"

We modified the text as suggested by the reviewer. For example the relevant subsection of the Results and Discussion has been retitled "Is Acari Monophyletic?"

In detail, the authors have sampled about 20 mites which is far less than, say, the ca. 90 ingroup mites of the Pepato & Klimov (2015) study or the 142 ingroup mites sampled by Dabert et al. (2016); both of whom did NOT recover a monophyletic Acari!

I'm not disputing the present author's results, but I think they are being disingenuous to reject previous (molecular) studies which did not recover a monophyletic Acari without explaining clearly why their data, from a much more limited sample of taxa, is likely to be giving a more reliable result. Is the fact that they are using genomic-scale datasets the key difference here? Is there precedent for these methods being fundamentally better for phylogenetic reconstruction?

There are several reasons and they are all stressed in our paper: (1) We use genome scale datasets and (2) very importantly, we strived to limit the potential recovery of artifactual topologies caused by the presence of fast evolving genes and taxa (which are present in every phylogenomic dataset). Using a broad taxon sampling is important, but only to the extent that the taxon sampling does not result in the generation of datasets dominated by fast evolving taxa. A key aspect of our study is the inclusion of short-branched Acari. In our results both Parasitiformes and Acariformes include two sister clades of short and fast species. This heterogeneity of lineages within both Parasitiformes and Acariformes is a very strong indication that our taxon sampling was effective at breaking long branches (i.e. Acari is not just a long branch attraction artifact joining exclusively long branched species). Finally, (3) we used models that fit the data much better than those used in the previously cited studies (CAT-GTR+G and CAT-GTR+G with Dayhoff recoding).

In short, although we use fewer species than in the best (previous) molecular studies (which we cite), we draw on orders of magnitude more amino acids and used more sophisticated models of tree reconstruction that much more efficiently counter attraction artifacts and a balanced taxon sampling including short branched Acari. Overall, we do not think that we are disingenuous. It is accepted by the community that conclusions of previous (molecular but pre-genomic) studies that used datasets an order of magnitude smaller than ours need testing and cannot be considered conclusive.

I also note that two key mite groups (Opilioacarida and Holothyrida) were not sampled at all. Opilioacarids are important in that they are widely seen as 'primitive' mites retaining several plesiomorphic characters for arachnids. Holothyrids are interesting as the putative sister-group of the ticks. I realise that material of these rare mites may not have been available, but I would have expected the authors to at least note that these lineages could not be sampled and thus to be a little more circumspect in presenting their results.

We agree, these data are currently not available. We added an explicit statement that we were not able to sample Opilioacarida or Holothyrida.

Another key non-mite group which is missing here are the palpigrades. Often promoted as 'primitive' arachnids, in some phylogenies (e.g. van der Hammen 1989) they were interpreted as closely related to at least the acariform mites and are thus of some relevance to the hypotheses being tested here. The lack of data for this taxon should also be conceded/noted in the Results and Discussion and the implications of this considered.

We were not able to also include palpigrades, as the reviewer remarked. We added a sentence (see also previous comment) to the Discussion to stress that there are still limitation in taxon sampling.

I would also like to stress that the mites that are sampled are taxonomically very localised (three tetranychoids: two in the same genus *Tetranychus*), two species of the highly derived parasite *Varroa*, etc. In this context I don't think the manuscript really contributes much to the study of relationships within the mites (as per Suppl. Fig. 2). The phytoseiid/dermanyoid and the argasid/ixodid pairings are non-controversial and not actually very interesting here.

The referee is correct, however it should be noted that we have not emphasised the internal relationships much. We simply stated what the groups were.

The monophyly of Trombidiformes and Sarcoptiformes respectively is hardly being tested here with the eleven acariform terminal taxa used. Most of the major trombidiform lineages (Spaherolichida, Labidostomatides, Eupodides, Anystina, Parasitengona, Heterostigmata) are missing. In this context I think that calling this "...taxonomically well-sampled datasets..." in the Abstract (p. 1) is somewhat misleading.

We have removed this misleading sentence and we stress that our datasets have been specifically designed to counter systematic error.

Given that the main conclusion is that Acari is monophyletic, I would have expected a little more discussion about what the sister group of Acari is; particularly in the section 'Phylogenetic relationships within Arachnida'. Implicitly the authors recover (Acari+Solifugae), and might want to discuss whether this has any morphological support. As they note, several characters already support Acariformes+Solifugae, but do ALL mites and camel spiders have anything in common?

Off the top of my head, I can think of the epistomo-labral plate and the fact that the free finger of the chelicera articulates ventrally against the fixed finger; although these characters may also be present in pseudoscorpions as well, e.g.

Dunlop, J. A. 2000. The epistomo-labral plate and lateral lips in solifuges, pseudoscorpions and mites. In Gajdos, P. & Pekár, S. (eds). Proceedings of the 18th European Colloquium of Arachnology, Stará Lesná, 1999. Ekológia (Bratislava), 19 (supplement 3/2000): 67–78

We now discuss all the above and have added a short discussion of the epistomo-labral plate in this context based on the Dunlop (2000) paper, and also note the articulation of the cheliceral free finger, citing Dunlop and Alberti's (2008) Table 1, in which this is listed as shared by Solifugae and both subclades of Acari.

The authors do briefly discuss Opiliones + Ricinulei, although I feel that this is an unconvincing clade with little or no morphological support. The only thing in common I can think of is an elongate second pair of legs, but this is not present in the basal cyphophthalmid harvestmen and is probably an Ingroup character for Opiliones. That said, prior to the Hansen & Soerensen (1904) monograph "On two orders of Arachnida", it might be worth noting that ricinulids were traditionally regarded as unusual harvestmen.

This clade emerges in our Bayesian analyses, but we agree with the reviewer that the morphological support, as well as previous support based on molecular data, is almost nonexistent. Therefore, we take these results with caution (see text). We have added the point made by the referee that early taxonomists recognised a group consistent with this "clade". In response to referee 2 we also add a reference to a group of Opiliones, Ricinulei and Solifugae recovered by Sharma et al (2014) using their 500 slowest evolving genes.

In summary, I would recommend that the authors (a) more clearly acknowledge the gaps in their taxon sampling, (b) make clearer why their results from a more limited dataset are likely to be more reliable than those of previous studies and (c) discuss in more detail - preferably with some morphological characters - the relationships between the arachnid orders recovered in the present work; especially the putative sister group of their monophyletic Acari.

The revised text addresses these concerns by (a) stating that our arachnid taxon sampling is mostly complete (to our knowledge and as stressed by reviewer 2 the best sampled Phylogenomic dataset to date -), although not yet ideal, (b) that our results are based on the most evenly-sampled genomic dataset to date at the order level, and (c) adding more discussion of some of the key relationships in our trees based on morphological grounds, particularly concerning the monophyly of Acari and its sister group.

MINOR CORRECTIONS

p. 1: "hemophagic" or (p. 2) "haemophages"? Please check British versus US spelling and be consistent.

Done

p. 2: According to version 3 of "A Manual of Acaology" (Krantz & Walter, 2009) there are about 55,000 described mite species.

Done

p. 7: Supplementary Information better "...with focus on the..."

Done

p. 8: don't capitalise titles of papers, i.e. "A Review of the fossil record of spiders (Araneae) with special reference to Africa. and description of a new specimen..."

Done

p. 8/9: Giribet's 2018 paper has now been formally published with volume and page numbers of "237, 7-13".

Done

p. 9: Should it be "Sperling, E. A." [not E. a.]?

Done

p. 9: should it be "Lake, J. A." [not J. a.]?

Done

p. 10: "An asymptotic equivalence of choice model by cross-validation and Akaike's criterion"

Done

p. 10: "...relationship with Pseudoscorpiones (Arachnida)."

Done

p. 11: I think if you want to cite Weygoldt and Paulus's cladistic results you should really refer to the second part of their paper, i.e.

Weygoldt, P. & Paulus, H.F. Untersuchungen zur Morphologie, Taxonomie und Phylogenie der Chelicerata: II. Cladogramme und die Entfaltung der Chelicerata. Z. Zool. Syst. Evol. 17, 177–200 (1979).

Please note that when this paper was published the journal had a German title.

Done

p. 11: "Cross-validators choice and assessment of statistical predictions."

Done

MINOR CORRECTIONS: SUPPLEMENTARY DATA

p. 4: Italicise the journal name "Nat. Protoc."

Done

p. 5: I think the more usual Journal abbreviation is "Phil. Trans. R. Soc. B" REPLY:

Done

p. 5: "Improved modeling of compositional heterogeneity supports sponges as sister to all other animals.

Done

TABLE S1

Is there any particular reason why two Illumina sources (for *P. citri* & *R. robini*) are italicised.

Done

Tetranychus cinnabarinus [not *Cinnabarinus*]

Done

Phytoseiulus [not *Phytoseiulus*]

Done

I do not mind my identity as a reviewer being revealed

Jason A. Dunlop, Berlin

Reviewer #2 (Remarks to the Author):

Review Lozano-Fernandez et al.

In their manuscript entitled 'Genome-scale chelicerate datasets support the monophyly of mites and ticks, reconciling molecules and morphology' the authors describe a phylogenomic approach to address the evolutionary relationships among chelicerate orders. In contrast to previous studies with the same – or at least a similar – aim, Lozano-Fernandez et al. have compiled a data set of unprecedented comprehensiveness encompassing all but two of the known chelicerate orders as well as more slowly evolving representatives of the Acari. They separate their data into two different data matrices, one – data set A – encompassing 234 loci and 45,939 alignment columns, and the other – data set B – encompassing 3,982 loci and 34,839 alignment columns. On this basis, they then tackle the problem to resolve the backbone of the chelicerate phylogeny. Specifically, two main questions are addressed: (i) Do mites and ticks can be grouped into monophyletic Acari, and (ii) what is the phylogenetic placement of the horseshoe crabs. From their analysis, three results emerge. First, a well selected data set is necessary from which an interpretable phylogenetic signal can be extracted. Second, what is left on phylogenetic signal even in the hand-selected data requires Phylobayes in combination with the CAT-GTR-G model to be correctly interpreted. And third, a tree is presented that reflects what morphologists have claimed for quite some time: Mites and ticks are monophyletic, and horseshoe crabs form the sister of the terrestrial chelicerates suggesting only a single transition from aquatic to terrestrial life style.

In a strict sense, the manuscript provides a little less novelty than it seems on the first sight. Basically, the two methodological issues have been already extensively discussed in the literature, and this study adds no further points that could help to further shed light on this point. This leaves the insights into the chelicerate phylogeny as the main driving point. And indeed, the authors claim that they could clarify some of the controversies that exist in the literature with respect to the intraordinal relationships within the chelicerates. This reasoning is one of the biggest concerns I have with this manuscript. Reading the manuscript a bit naively, one may get the impression that the claim is based on eventually – and after many trials – observing a phylogenomic tree that is congruent with what morphological data told us for quite some time. Obviously, this would be poor reasoning because it would make the morphological evidences the gold standard. I, therefore, would wish for a more objective approach of the problem where the results of the phylogenomic analyses are presented in a way such that they can stand for themselves. Only then the comparison to the morphological data should be performed. If this can be achieved, the phylogenomics studies can then indeed provide another line of evidence that help in resolving the early stages of chelicerate evolution.

We agree with the reviewer that the main point of the paper is the chelicerate phylogeny. We also agree that using morphology as a gold standard would be unsound reasoning, but we would like to stress that this is not

the basis for our reasoning. The strength of our argument is not based on morphology being a gold standard, but on the fact that Consilience (Whewell 1840) of hypotheses is the only epistemologically sound tool to corroborate hypotheses and develop theories.

Consilience (i.e. convergence) of different lines of evidence on the same result is what makes evolution a theory (McInerney O'Connell and Pisani 2015 Nat Rev Microbiol), and we have championed the application of Consilience as a guideline to arbitrate between phylogenetic disputes for more than a decade (see for example Pisani et al. (2007) Acta Biotheoretica; Rota-Stabelli et al. (2011) Proc R Soc B; Campbell et al. (2011) PNAS; Pisani et al. (2015) PNAS; McInerney et al. (2015) Nat Rev Microbiol; Pett et al. (2018) BioRxiv).

The present paper uses the same well developed and philosophically robust framework to attempt discriminating between alternative phylogenetic hypotheses.

That said, we do agree with the reviewer that our genomic analyses should stand on their own and it is our opinion that they do, as our results are logically consistent and are also consistent with the results of the new analyses that we performed as suggested by the reviewer.

Major issues

1. The authors present two data sets of which they consider one, data set A, superior to the other. I think having two non-overlapping data sets telling a consistent evolutionary story would be a relevant asset to this study. It is therefore sad that data set B earns very little attention and no attempt is made to improve it over what was achieved with the standard procedures.

We agree that having two independent dataset is a significant asset and below we address the concerns of the reviewer.

Specifically I wonder

a. What is the overlap between data sets A and B. Somehow I can imagine that data set A should be a subset of data set B. If so, this overlap should be removed to have two independent views on the same process. If they do not overlap with respect to the represented loci, I would be highly interested why this should not be the case.

This is an excellent point, and we have to admit that we did not consider it in the original submission. In fact, Matrix A is indeed a subset of Matrix B. We agree with the reviewer that having a Matrix C (with C=Genes in B - Genes in A) would allow for useful comparison between A and C (which will be fully independent) and we have now assembled and analysed a new dataset (named "C" – for consistency) that does not overlap with A.

Results of these analyses have been integrated in the text and figures.

We thank the reviewer for pointing this out as we think the inclusion of a Matrix C that is fully independent from A has significantly improved the paper.

We have added a saturation plot for Matrix C to the main figure (Fig 1) Matrix C, indicating that it is also more saturated than Matrix A and comparable to Matrix B, as well as a ML tree obtained from the analysis of Matrix C. Note that it makes perfect sense that Matrix C is approximately as saturated as B as they are very similar to the sites in B that are from A are an overall minority.

Note that the Bayesian analysis of Matrix C failed to reach convergence (hence we ended up presenting only the ML tree for that matrix). We suggest that this is most likely because by removing the genes in Matrix A the signal to noise ratio of the dataset deteriorated, consistent with our opinion that the genes in A carry a higher amount of *bona fide* phylogenetic signal than those that are not in A. Note also that as one would have expected, in all occasions where Matrices A and B disagreed, Matrix C was consistent with Matrix B, suggesting that nodes that emerge only in B and C are likely to be tree reconstruction artifacts as Matrices B and C are more saturated than is Matrix A.

b. The authors mention the higher level of saturation in data set B compared to A and explain with this the poor performance in the tree reconstruction. Looking at figure 1A, I wonder why no attempt was done to decrease the level of saturation in this data set by removing those genes that are particularly effected. The authors even write in the manuscript that no attempt was done to do so. I am a bit puzzled that – given the severity of the reconstruction problem – not all attempts are done to maximize the information content in the data. As already stated above, it would be highly desirable to have a consistency criterion at hand that can be used to generate confidence in the phylogenomic reconstruction independent from morphological evidences.

First we would like to note that we do not discuss the alternative matrix as having different level of performance, as that would indicate that we had a predefined idea of the results we should have obtained. We discuss differences and similarities in our trees with reference to the saturation in the data and use the level of saturation to decide what topology should be more trustworthy. We then move on to observe that, in our opinion unsurprisingly, the least saturated and hence (to our opinion) more reliable dataset is also the one that reaches the greater level of consilience against an independent source of evidence. This is as one would expect only if an underpinning species history shaped the evolution of both genomes and morphology.

Having clarified this, our saturation plot analyses show that the genes in Matrix A are much less saturated than those in C (the complement of A; $B=A+C$). Hence, we literally did the experiment suggested by the reviewer by independently analysing Matrix A. In our original write-up we did not include Matrix C. However, thanks to the new analyses we performed, as suggested by the reviewer, we now have also the result of the “counterexperiment” (a saturation plot of Matrix B to the exclusion of the genes in A), which show that the genes in C are indeed more saturated than those in A and that therefore Matrix B to the exclusion of its faster evolving genes is well approximated by Matrix A.

Note that we discussed the low rate of evolution in the genes in Matrix A in a different context in Pisani et al. (2015) PNAS.

c. I am wondering about the nature of data set B. The authors state that they considered almost 4,000 loci (orthologous groups) in this data matrix. Still, the alignment is not even 35,000 amino acids long. Thus, the average contribution per locus is less than 10 amino acids. I think this clearly shows that there is ample room for the improvement of this data set. Note, the data set A comprises only a little more than 200 genes but is about 10,000 amino acids longer.

Matrix B has been optimised by stringently pruning it in order to reduce the level of missing data (less than 14%), in order that all taxa and amino acid position contain maximum levels of information. Matrix A has been optimized to include genes that are single-copy and slowly evolving, in order to reduce reconstruction artifacts. We could have analysed the data with all the gaps included, but this would have caused problems of computational intractability while not improving the quality of the inference. Too many gaps are deleterious to the analyses.

d. The ‘orthology’ assignment procedure used in this study is an interesting approach that requires a massive amount of human interaction. I am wondering why not a published approach for an automated extension of orthologous groups with further sequences from transcript data was applied. Moreover, the authors state in the supplement that they visually inspected each tree and each candidate ortholog in order to make the decision to keep a candidate, or not. Somehow I cannot believe that this procedure was done for all 3,982 orthologous groups that are considered in data matrix B. Lastly, how can one rule out that this visual inspection and sequence selection does not introduce an unwanted bias by favoring those orthologs that place a taxon at a certain position?

This is an important comment that requires an answer developed across multiple points.

- (1) There are many automated approaches for orthology assignment all of which are to some extent arbitrary. While we think that OMA and OrthoMCL are probably the best ones currently available, none of them is ideal and they all have their drawbacks. Our pipeline, MoSuMa, is not an alternative to OrthoMCL or OMA. Rather, it is a tool that can be used in association with OMA and OrthoMCL to increase their efficiency.
- (2) Our Pipeline (MoSuMa) is published (Lozano-Fernandez et al. (2016) Phil Trans R Soc, and after that by Tanner et al. (2017) Proc R. Soc. B.) and publicly available (Link in text) – We have modified the text to made this clear as it was not clear before. MoSuMa does not do the same job that OMA or OrthoMCL do. OMA and OrthoMCL generate a dataset *denovo*. Differently, the remit of MoSuMa is to expand an

already available dataset (e.g. a dataset generated with OMA for a smaller number of taxa) or a legacy dataset like our Matrix A. MoSuMa is faster than OMA to add sequences to a pre-existing dataset and that, we hope, will make it appealing and a useful tool to use in combination with OMA or OrthoMCL. Indeed, one cannot use MoSuMa from scratch, but if you have a seed dataset composed of a set of protein families identified, for example, using OMA over 23 transcriptomes (as in our case), MoSuMa can expand that dataset to, say, 95 species (as in our study) in a fraction of the time that OMA would require.

- (3) MoSuMa does not involve human interaction. It uses well-characterised rules that do not impinge on an orthologue to be in any specific position within the ingroup. Indeed, only putative orthologs nesting within the outgroups are rejected; while if more than one putative orthologue is identified for an orthogroup, MoSuMa uses rules based on statistical tests performed on branch-lengths to discriminate between them. This is based on the idea that paralogs must have longer branch lengths than the orthologue as they diverged from a previous gene duplication. These rules were described in detail Lozano Fernandez et al. (Phil Trans 2016), and it is our opinion that they validly identify the correct orthologues with a level of accuracy comparable to that of OMA (see below our answer to the next comment of the reviewer – suggesting a comparison between the results of our pipeline and those of OMA). We have modified the text to make it clear that MoSuMa is fully automated. The previous text failed to clarify this as we explained the steps carried out by MoSuMa without pointing out that such steps are automated in our pipeline.
- (4) However, we agree that our legacy dataset (Matrix A) was initially assembled using approaches that required significant human intervention (see Rota-Stabelli et al. 2011; Campbell et al. 2011). The inclusion of a new OMA-based dataset (Matrix B, and now C as well) was investigated exactly to make sure that subjective choices that might have been made when assembling the backbone of Matrix A (in Rota-Stabelli et al. 2011) did not induce unwanted biases in our analyses. The analyses we performed here, suggest that that was not the case.
- (5) Gene trees were visually inspected only for Matrix A, after the orthologue was selected by MoSuMa, and only to make sure that there was no obvious problem with our pipeline. For example, if all the orthologues we added using MoSuMa ended up clustering with the outgroups and having exceedingly long branches it would have indicated a failure of the pipeline (perhaps the presence of a bug). Nothing of the like was identified. The added orthologues showed branch lengths generally comparable with those of the other genes in the dataset and were broadly distributed within the ingroup as one would expect from gene trees that frequently do not have the resolving power to cluster sequences with confidence along any specific branch of the gene tree.
- (6) The reviewer is correct that the trees were only manually inspected in the case of dataset A. For Matrix B this would have taken far too long, and would not have been necessary as tests performed on Matrix A showed that MoSuMa behaved as expected. We modified the text to make this clear.
- (7) We further tested the accuracy of OMA orthology prediction by comparing the genes over which Matrix A (MoSuMa) and B (OMA) overlap (see below and in Supplementary Information), finding that both methods in the largest majority of the cases retrieve the same orthologue.

We have modified the text and added a new section in Sup Info (see next comment) to make all above points clearer.

e. The data matrix B was compiled using an integrated approach that combines OMA and the authors' own custom approach for ortholog search. This combination was apparently necessary to cope with the computational burden. Two questions arise: First, I trust that in some cases there should be ortholog assignments available both from OMA and from the "BLAST-based approach". It would be interesting to know in how many cases they differ. Second, the OMA approach requires a subselection of taxa from which the orthologous groups were collected. How were these taxa selected?

This is a very good suggestion and we thank the reviewer for this.

We compared the differences in the orthology-assignments between OMA and MoSuMa by analysing a subset of the orthology assignments. Specifically, we selected the 10 orthogroups shared between Matrices A and B that had the highest and the lowest gene coverage in the OMA orthology assignment approach (which used a subselection of 23 taxa - more below). This experiment is described in the revised Supplementary Methods (text reported here as well). It showed that MoSuMa and OMA almost always retrieve the same orthologues, or one of the two methods find an orthologue while the other fails to do so. Overall this is reassuring as it is conservatively indicating that when they both find an orthologue they almost invariably agree.

This is the key section of the new text from the Supplementary Information: "...For the genes with best coverage, we found that for the 230 cases analysed (inspecting the 23 taxa used in the OMA analyses that are also in Matrix A), OMA and MoSuMa retrieved identical orthologous genes for each species in 82% of cases. In 17% of the cases, only one of the methods retrieved an ortholog. Only in 0.8% of cases (2 instances out of 230) did OMA and MoSuMa disagree on the protein assigned as the ortholog. We repeated this analysis with the 10 orthogroups shared between Matrices A (MoSuMa) and B (OMA) that had the lowest gene coverage in the OMA orthology assignment approach, finding similar results. From the 230 cases analysed, in 70% of the cases both methods retrieved the same ortholog (or absence of it), and in 30% of the cases, only one of the two methods retrieved an ortholog. When using the 10 orthogroups with the lowest OMA coverage, not a single case was identified where the two methods disagree (See supplementary Table 3)".

The subselected taxa for the OMA analysis were based on the relative abundance of each of the orders in the final dataset, and the specific selected taxa were based on species presenting high-quality transcriptomes. We included one species for each outgroup clade, one species for each order with less than five taxa included, two species for orders containing between five and 10 taxa, and three species for orders with more than 10 taxa.

We corrected the text that erroneously stated that we used one species for each chelicerate order, and in Supplementary Table 1 all the species subselected for OMA are marked in bold.

2. In their phylogenomic analysis, the authors find at least one grouping that is consistently observed across all trees. Most prominently, this is the alliance of Opiliones and Ricinulei in a sister clade. Somehow, I have the impression that this true novelty earns surprisingly little attention in the discussion. One even gets the impression that the authors are not confident that this grouping is not an artefact, which probably is reflected in the sentence that these groupings require 'corroboration'. If anything, I would think that this study – if the authors consider it comprehensive – should provide the basis for coming up with new hypotheses.

We expanded the discussion of this grouping, but as this is a clade that has not previously been retrieved either based on morphological or molecular grounds, we are not particularly confident about its validity. We agree with the reviewer it is an interesting finding, but one that can only be reported as an hypothesis that will need to be tested by further studies. We added a mention however that a grouping of Opiliones, Ricinulei and Solifugae was recovered using a large set of the slowest evolving genes in a previous study (Sharma et al., 2014). We also indicate some morphological support for that group.

In addition I see a number of minor yet essential issues.

1. The authors mention that they, in the ML framework, used LG+G+I as a model for sequence evolution. Could it be true that the F parameter for estimating the amino acid frequencies from the data was not suggested when the model with the best fit was searched with ModelFinder?

Thanks for the suggestion. We double checked the model chosen by ModelFinder, and in Matrices A and B all the information criteria available in ModelFinder chose LG+I+G4 as the best fitting, and in the case of matrix C the F parameter was chosen LG+F+I+G4. The new text now describing Matrix C as well points this out.

2. The authors extracted at least part of their data from transcriptome reconstructions. Although I know that the terminology is not strict in this context, I think it is worthwhile making a distinction between gene sets (called "proteomes" now and then) inferred from whole genome sequences, and those sets that have been inferred from transcriptomes. Reason is, we will – by necessity – miss always very lowly expressed genes and those that are not transcribed at all in our transcriptomes. Unfortunately, it is one of the inherent assumption of the ortholog search tools, such as OMA, that they have the full gene set at hand for the inference.

Thanks for pointing this out. We agree with the reviewer and acknowledge this point, and we tried to be consistent throughout the text and find a constant name for those proteins derived from a transcriptome (on which our work is mostly based), naming them as an effective set of translated proteins.

3. On page 4, the authors state that their cross validation test revealed that the LG+G+I was not the overall best fitting model and refer to something that was written further up in the text. I find further up no further mentioning of the cross validation in the context of the LG model. This should be clarified. Furthermore, it might be worthwhile to be a bit more precise in the wording as also readers not professionals in phylogeny reconstruction in a likelihood framework should be able to follow the argumentation.

We have tried to be more detailed and clear in the text. We performed cross-validation analyses only for our least saturated dataset (Matrix A) to assess model fit, as described in the PhyloBayes manual. CAT-GTR+G emerged as best fit with GTR+G as a distance second best (this was all expected based on previous analyses and theoretical argument about the need of modelling site-specific and lineage specific compositional heterogeneity – see Feuda et al. (2017) for example). For the IQTree ML analyses we could not use the overall best fitting model (CAT-GTR+G) because this model cannot be implemented in a ML framework (only in a Bayesian framework). Accordingly, for the ML analyses we used the best fitting among the models suggested by ModelFinder in IQTree. These were LG+I+G4 (Matrix A and B) and LG+F+I+G4 (Matrix C). However, we already knew, from the Bayesian cross-validation, that if ModelTest could consider CAT-GTR+G it would have been selected as best fit - Model selection is always relative to the models that can be tested. The fact that we could not implement the overall best fit models in our ML analyses is the underpinning reason for our arguments that when the Bayesian and the ML results disagree we should infer the Bayesian more likely to be correct. It is not an argument about the methodology itself (i.e. it is not a question of preferring Bayesian analysis to ML), but a question of preferring the objectively best-fitting model (CAT-GTR+G) that is only implemented in a Bayesian framework.

4. In the methods, the authors explain why they used *D. pulex* sequences as a ‘bites’.

a. Probably most of the readers will not be familiar with the custom approach used here to predict orthologs in transcript data. Hence, they will not be familiar with the particular terminology, such as ‘bites’.

We modified the text and removed this term, and instead stated that: ‘sequences from *Daphnia pulex* were used as BLAST reference seeds. *D. pulex* was chosen as our reference species because it has full coverage for the 233 orthologs in our original^{2,14} dataset’. We hope this clarifies.

b. *D. pulex* was used because it has “phylogenetic proximity” to the chelicerates. I would appreciate if this could be elaborated a bit further. *D. melanogaster*, just as any other insect, should have the same level of “phylogenetic proximity”, is this correct? But probably this is not what the authors have in mind?

The reviewer is correct, and the phylogenetic distance between *D. melanogaster* and *D. pulex* is equivalent respect chelicerates because both share a last common pancrustacean ancestor. Both organisms are model systems in arthropods and their genome sequence is very complete, but we have chosen specifically *Daphnia pulex* because, with reference to Matrix A, this species possess the full coverage of the consider orthologs while *D. melanogaster* does not. Hence if we were not to use *D. pulex* we would have had to use more than one species to seed our searches, which would have made our methodology less tidy.

Note that this is discussed in the methods and we tried to amend the text to make it clearer, see our answer to the previous comment.

5. Figure 1A – the caption is a bit sparse. What do the individual dots represent? Not every reader is familiar with such plots.

We have extended this caption to indicate the interpretation of the plot and that the dots represent the intersection between branch lengths obtained using uncorrected genetic distances compared to the patristic distances derived using LG+I+G4 (or LG+F+I+G4) for each pair of the 95 terminal nodes. Note that as the Bayesian analysis of Matrix C did not converge, we have changed these plots from the first submission. Initially we compared distances derived from the Bayesian trees obtained under CAT-GTR+G against raw uncorrected distances. Now we compare distances from the ML trees (again against the raw observed distances). It is important to point out that for saturation plots it is not necessary to use the overall best fit model because any model will identify a proportion of the multiple substitutions that the raw, uncorrected distances miss. Hence while under CAT-GTR+G the pattern should be more striking (i.e. the regression statistics would be worse), the relation between the slopes of the regression lines is expected to change in a correlated way. In contrast, presenting Sat Plots under CAT-GTR+G for Matrices A and B but under ML (LG+F+G+I) for dataset C would

have biased our results. This is because LG would have identified less multiple substitutions in C than CAT-GTR+G would have found in A and B, potentially misleading our conclusions.

So in short, any model is fine for doing saturation plots across multiple datasets, but using models that significantly differ in their fit to the data over different datasets would bias the results against the dataset that was analysed using the best fitting model (in our case using CAT-GTR+G for Matrices A and B and LG for Matrix C would have led us to think that C was less saturated than it actually is).

Figure 1B-D. There a quite a number of branches without support values. If this is maximal support, then this should be mentioned in the legend.

Yes, they represent maximal support. We stated that lack of a number indicates maximum support.

7. Figure 2 – The legend is very specific with respect to details of the Bayesian analysis. I trust that quite a bit of this can go into the methods section (Burn-in, total cycles, subsampling frequency).

We have moved the details of each Bayesian run to the Supplementary extended methods section.

8. Although the analysis focusses on the interorder relationships, I wonder to what extent the topologies within the individual orders agree (or disagree) between the different trees. Somehow one would expect that these are a bit less problematic to resolve, and hence we would expect the trees to be largely congruent. Is this indeed the case? If not, what would be a possible explanation for any incongruence?

Although we haven't focused on the intraordinal relationships, they are almost equivalent between analyses. The Supplementary Data depict a tree with classification of Acari on it, demonstrating that clades within that group are ones recognised previously by taxonomists. The same applies to the other two large groups included in the study, Opiliones and Araneae. Groupings are taxonomically familiar.

Reviewer #3 (Remarks to the Author):

In the present manuscript Lozano-Fernandez and co-authors study the relationships of some (most) chelicerate lineages using primarily available transcriptomic datasets. Data for additional 4 taxa is generated as part of this study. Although the taxon sample is rather wide and the first part of the title is more general, the authors mainly focus on one particular results, which is the monophyly of acari (a lineage that includes both, mites and ticks). That is indeed an interesting finding given the general lack of support of monophyletic acari based on molecular data although there are numerous morphological similarities that have traditionally supported this group (as mentioned by the authors).

We agree that there is substantial morphological similarity and that despite the differences between Parasitiformes and Acariformes pointed out by reviewer 1 (and painstakingly catalogued by Gerd Alberti in published work), there still is more morphological similarity between these two lineages than between each of them and other chelicerate lineages. As pointed out when discussing the comments of reviewer #2, we do not take morphology as a gold standard, but neither do we take genomics as a gold standard. We take as our guiding principle the convergence of independent lines of evidence on specific hypotheses (Consilience) and consilience of morphology and genomics on monophyletic Acari is indeed very interesting. As we point out in replies to reviewer 2, it is Consilience that changes hypotheses into a theories.

Few other results are also mentioned briefly (e.g., the sister group relationship between Opiliones and Ricinulei) these are not discussed much.

The reason for the lesser focus on these relationships is because they are at this stage not supported by consilience of independent data sets. Hence, while we discuss them and investigate possible synapomorphies for these groups, we do not think our evidence is in itself sufficient to claim they are anything more than hypotheses in need of further testing.

Although the finding of monophyletic Acari based on molecular data is an intriguing result I have several concerns regarding the ms. These are listed here in no particular order.

I guess my main concern is the supposed high support for acari and the overall novelty and significance of this finding. This clades is indeed present in the results but support is no that high in several cases (e.g., Fig 1B, D). When support values are not shown I assume that means full support (100%) but this is not stated anywhere explicitly but 0.68 is not high and 0.74 can be considered but I would not call it high. Furthermore, the authors state that they have checked Phylobayes runs for convergence and ESS and have made sure that these are within the acceptable limits (ESS>50 and maxdiff values <0.3 which is the bare minimum) but they do not provide the statistics that were found for their analyses. So if ESS were rather close to 50 and maxdiff to 0.3 I would strongly suggests that the authors make an effort to improve these. In any case, I think that ESS, maxdiff and other similar statistics should be provided (e.g. in a suplemenry table, text etc.).

We clarified the figure legends to make sure that it is now clear that only support values different from (1 or 100%) are reported. Support for Acari is strong in all analyses, including all the three ML analyses, (Matrix A = 97, Matrix B = 93 and Matrix C = 97), but the Dayhoff recoding (0.68), where the support is moderate. Dayhoff recoding can, however, induce some loss of signal (see also Feuda et al. 2017) and if one takes that into account the conclusion is that overall the support for this node from our dataset is indeed very strong.

As for the convergence statistics, we agree with the reviewer, and we have added them for all Bayesian analyses now in the supplementary extended methods section. A note however, is that convergence statistics are just guidelines and not to be taken as rules of thumb. Convergence statistics are good for the analyses of Matrix A, but less so for those of Matrix B, although for this matrix they are still acceptable. In contrast, Bayesian analyses of Matrix C did not converge hence we report only ML results for this dataset. Note that the decreasing level of convergence achievable from Matrix A to B to C is to our opinion logical and was to be expected, as it reflects the quality of the signal in the three datasets (i.e., A good, B less so, C rather poor as it excludes the signal from the less saturated genes from Matrix A).

The authors also do not provide much discussion on what previous molecular studies have found regarding acari and how this compares with the results presented here. What is the broad impact of this finding? Why it is important in a boarder context? Nature Communications is a multidisciplinary journal with a extensive readership so putting the present findings in a broader context is very relevant here. What about previous findings on the internal structure of the two main linages within acari? This is basically missing in the discussion.

Acari are of tremendous biomedical and biotechnological relevance. In addition, they are of significant ecological importance and from the evolutionary perspective, whether Acari is monophyletic or not has an effect on our understanding of chelicerate diversity and global biodiversity. Indeed, monophyletic Acari is more species-rich than spiders thus providing a significantly different view of what the Chelicerata biodiversity looks like. We have emphasised the question of the monophyly (or diphyly) of Acari rather than their internal relationships because our taxonomic sampling is designed around that test. As our figures indicate (by being able to label the groupings with widely accepted clade names), we recover standard internal groupings within the two subclades of Acari

We have tried to amend the text to make the relevance of our findings more obvious.

The use of the word diversification in the third subheading of the results is not appropriate. Relationships are discussed in this section but not diversification patterns. There are no diversification analyses either.

We have changed the name of this section, as the reviewer is right that no diversification analyses were done.

There is a mention of what morphological data has suggested in terms of the Opiliones and Ricinulei but what about molecular data? Previous analyses based on molecular data put Opiliones in the vicinity of Ricinulei so that is not that unexpected finding here. I think that some more discussion and references are needed here.

Some molecular datasets from Sharma et al. 2014 did put Opiliones in the vicinity of Ricinulei but sun spiders were in between. Ricinulei + Opiliones in isolation never appeared as sister. We have expanded the discussion to address the comment of the reviewer and we now also cite the paper of Sharma and collaborators in this context.

When the authors talk about the use of models in the ML analyses (last lines of page 4) they use the term "overall best fitting model". What "overall best fitting model" means? You state that LG+G+I was the best-fit model selected with ModelFinder for the ML analyses some lines above yet it is not "overall best fitting" Further, you state "we suggest that when results from our LG+G+I and CAT-GTR+G analyses (with and without Dayhoff recoding) disagree, the latter should be considered more reliable." What would be the rationale for that? Selecting best fit models is based on a quantitative approach (you have used ModelFinder). Here you suggest that when model selection finds a model that results in a topology that does not fit a "preferred" case (as in the LG+G+I analyses) model that leads to results that comply with this "preferred" topology should be favored.

But what is the objective quantitative criterion that you suggest? I think that the statements made in this sentence need to be explained better and with more clarity. As it is, I do not find this very convincing.

We have now clarified the text (see also our answer to Reviewer N.2). The "overall best fit model" is CAT-GTR+G. We performed tests (Bayesian cross validation that showed this to be the case). LG-based models are suboptimal in their fit to our data when compared to CAT-GTR+G. However, there is no ML implementation of CAT-GTR+G. Hence for our ML analyses we could not use the overall best fit model. Given that CAT-GTR+G could not be used for our ML analyses, we performed a second model selection test to compare the various models implemented in IQTree (this test did not include CAT-GTR+G as it is not implemented in IQTree). These tests identified LG-based models as best fitting the data (but only with reference to the set of models implemented in IQTree, and CAT-GTR+G would have emerged on top if it was tested). Hence for the ML analyses we had to use a model that, while best fitting within the context of the models implemented in the software we used, was suboptimal with reference to the complete set of models we used.

This is why we prefer results of CAT-GTR+G analyses to results of analyses derived using LG. Our rationale is based on the order of fit of models to the data, and as such it is robust and "the standard". Our argument has nothing to do with the topology we obtained in the various phylogenetic analyses.

We have clarified the text to make this clearer.

On Figure 1 A It looks like the low R square for matrix B is driven by relatively low number of outliers. I wonder how results might look if these positions are excluded and then Matrix B is used. In such way the dataset would be still larger than the matrix A and would not be subjected to an a priori data selection process.

Dots in the figure represent taxon-to-taxon distances (that is, the sum of branch lengths in the path from one species to any other). They do not represent sites. The specific dots to which the reviewer refers to are the longest branched taxa in the dataset, but they are also taxa of critical interest to the study and thus cannot be excluded.

Note that we have repeated the Saturation Plots (see above answers to comments to reviewer 2). These taxa still emerge as outliers but the charts look now slightly different. Irrespective of that, the overall conclusions we reached from the saturation plots is unchanged (Matrix A is less saturated than Matrix B and C).

Finally, text labels (e.g., tip labels) on supplementary figures hard to read.

We have increased the size of the fonts on the supplementary figures

The supplementary text in places need closer revision of style/grammar.

We revised the text and hopefully caught all errors.

In conclusion, I think that the novelty and the broader significance of the purportedly novel finding of monophyletic Acari (based on molecular data) are not well presented and discussed. I also have some concerns regarding the level of support for Acari and the convergence/ESS for the Bayesian analyses. Thus I can not recommend publication of this ms in its current shape.

However, I think that the presented results are interesting and there is a potential, after a proper review, for this ms to fit the type of studies that Nature Communications aims at. Thus, my recommendation is rejection with invitation to resubmit a revised version.

We have now addressed the concerns of all referees and added new analyses (based on Reviewer #2's comments) that we think significantly strengthen our arguments in favour of the monophyly of Acari (fully or moderately supported in all Bayesian and ML analyses). We hope the reviewers will find our new MS to have improved.

Reviewers' comments:

Reviewer #1 (Remarks to the Author):

The revised version of this manuscript has made some effort to address the previous points raised, has acknowledged the fact that some taxa could not be sampled and is now written in a way which is more circumspect in its conclusions.

I think it is now acceptable for publication and will represent a valuable contribution towards our understanding of arachnid phylogeny.

Jason Dunlop

Reviewer #2 (Remarks to the Author):

The revised version of this manuscript has addressed most of my initial concerns. I welcome the inclusion of the third data matrix C. It is, however, a bit unfortunate that the Bayesian tree search did not converge for this data set. In essence, the idea of having two independent data sets that both, given their level of saturation, should be equally suitable to address the phylogenetic problem under consideration, could not be realized. I leave it to the discretion of the authors to briefly discuss this aspect in the manuscript. However, I still see one major and a couple of minor issues that should be addressed prior to publication.

Major Issues

1. I still have problems with the fact that data set B, which should summarize information from almost 4,000 genes, is about 11,000 positions shorter than that of data set A, which comprises only 233 genes. The authors explain that this is due to the stringent post-processing of the data, and indeed the parameter values for Gblocks differ. I will now try explaining the reason for my concern a bit more in detail: The genes in data set A are almost all represented in data set B, this has become clear by now. While these genes contribute 45,000 alignment positions to set A, the more stringent filtering in B has removed 90% of this data, and now these genes contribute only about 4,300 positions to the alignment. This is now an indication to me that the authors are a bit too happy in removing data from data set B, for which they already have shown that it is phylogenetically informative. I wonder how this affects the analyses, and in particular the phylogenetic information content in data set B/C.

Minor issues

1. The authors state that the genes in data set A are less saturated than those in data sets B, and figure 1A seems to support this. Now, my naïve expectation would be the following: If I remove the data of all genes in data set B, which make up the less saturated data set A, then I would assume that the saturation would increase. However, figure 1A tells a different story. I have the feeling that this requires at least some attention (This is tightly linked to major issue 1).
2. Abstract – I suggest to remove the redundancy in the phrase “alternative, conflicting phylogenetic relationships”. Alternative relationships are always conflicting, and vice versa. Likewise, in the Results and Discussion section, I suggest to simplify the phrase ‘common set of shared genes’, which is equally redundant.
3. Page 3 – why is there a ‘~’ in front of the 40%? Probably it would be better to give the minimum number of species a gene has to be represented in rather than a percentage.
4. In data sets B/C, the authors have retained, prior to the Gblocks filtering, 3,982 genes that are represented in at least 40% of the 95 species. Given the Gblocks parameter settings, does a gene that is represented in say half of the 95 species have a chance to survive the filtering? Somehow, and given that

the resulting data matrix is 86.4% complete, I trust that this is not the case. I think this should be clarified. I am convinced that data sets B/C contain, after filtering, data from substantially fewer than 3,982 genes.

5. In the Results and Discussion section, the length of data sets A and B is specified, but not of data set C. Please add this information.

6. I have some difficulties with the statement that CAT-GTR+G model is the optimal one, whereas LG and its variants are referred to as 'sub-optimal'. I think we are all aware that all models that we have at hand are 'sub-optimal', as we can be sure that they do not accurately model the true evolutionary process. In essence, from all models at hand for this analysis, CAT-GTR+G gives the best fit to the data. Still, when compared to an unknown 'true model' it can be (almost) as wrong as the LG model. This should be somehow reflected in the wording.

7. Results and Discussion – The authors state that the Dayhoff recoded matrix confirms the support for a sister group relationship between horseshoe crab and Arachnida. Figure 1d seems to not support this statement since the split the authors refer to is not resolved. Instead we see a trifurcation.

8. The section "Phylogenetic relationships within the Arachnida repeats the findings from the previous section when stating "all analyses recover monophyletic Acari". I leave it to the discretion of the authors to remove this redundancy.

9. The authors state that the Dayhoff recoded matrix A does not recover the grouping of Acari and Solifugae. They interpret this as a suggestion that this grouping might be an artefact. I would not support this interpretation for two reasons: (i) the split uniting Solifugae and Acari in Figure 1A is supported with a posterior probability of only 0.77 anyway. So, one should be careful with referring this to a clade anyway. (ii) The support for the placement of Solifugae in Figure 1d is only 0.64, so it is essentially unresolved. Thus, the recoded data matrix A seems to not contain sufficient signal to suggest anything with respect to whether or not Solifugae are sister to the Acari, or not.

10. The authors write 'ambiguous support' when referring to the outcome of the ML analysis concerning the placement of Ricinulei and Solifugae. What is this 'ambiguous support' exactly, low bootstrap values? If so, why did they not use the same phrase in the context of low posterior probabilities?

11. What does it mean when "groupings are taxonomically familiar"?

12. Methods – what is an "effective set" of translated proteins?

In summary, I hope that my comments help to further improve the manuscript.

Sincerely,

Ingo Ebersberger

Reviewer #3 (Remarks to the Author):

In their revised ms Lozano-Fernandez and co-authors have introduced number of changes following mine suggestions and those of other two reviewers. I find the revised version greatly improved. Some parts that were unclear in the original submission (e.g. model selection) are now better explained and I find the authors' reply to my comments and the corresponding changes in the manuscript satisfactory.

I felt that it may be a bit odd to use "rock record" to refer to the fossil record but that is just because fossil record is more commonly used. Both are correct indeed.

This is an interesting and important contribution to our knowledge of chelicerae relationships and particularly mites and ticks and I am happy to recommend publication of the ms.

Reviewers' comments:

Reviewer #1 (Remarks to the Author):

The revised version of this manuscript has made some effort to address the previous points raised, has acknowledged the fact that some taxa could not be sampled and is now written in a way which is more circumspect in its conclusions.

I think it is now acceptable for publication and will represent a valuable contribution towards our understanding of arachnid phylogeny.

No Action to be taken.

We thank the reviewer for these positive comments.

Jason Dunlop

Reviewer #2 (Remarks to the Author):

The revised version of this manuscript has addressed most of my initial concerns. I welcome the inclusion of the third data matrix C. It is, however, a bit unfortunate that the Bayesian tree search did not converge for this data set. In essence, the idea of having two independent data sets that both, given their level of saturation, should be equally suitable to address the phylogenetic problem under consideration, could not be realized.

I leave it to the discretion of the authors to briefly discuss this aspect in the manuscript. However, I still see one major and a couple of minor issues that should be addressed prior to publication.

It is incorrect to state that the use of two independent datasets could not be realised as we present the results of the ML analyses of dataset C, which we were able to complete, and it is fully independent from Dataset A. It is correct that the Bayesian analysis of Dataset C could not be completed, as it failed to converge. However, in our opinion this is not unsurprising as the genes in Dataset A have less saturation and are more informative (as emerges clearly from the saturation plot analyses), and this is important to allow good convergence in Bayesian analyses under CAT-GTR. Indeed, starting with our initial submission we have stressed that dataset A is more reliable (based on saturation plots) than datasets B (and therefore and C). The latest result (lack of convergence in dataset C) confirms our opinion.

Irrespective of that, MLs infer a tree for dataset C that is no different from that of dataset B with reference to our key result (monophyly of Acari). Accordingly, our key result is supported by two independent datasets (A and C), irrespective of their relative quality.

Major Issues

1. I still have problems with the fact that data set B, which should summarize information from almost 4,000 genes, is about 11,000 positions shorter than that of data set A, which comprises only 233 genes. The authors explain that this is due to the stringent post-processing of the data, and indeed the parameter values for Gblocks differ. I will now try explaining the reason for my concern a bit more in detail: The genes in data set A are almost all represented in data set B, this has become clear by now. While these genes contribute 45,000 alignment positions to set A, the more stringent filtering in B has removed 90% of this data, and now these genes contribute only about 4,300 positions to the alignment. This is now an indication to me that the authors are a bit too happy in removing data from data set B, for which they already have shown that it is

phylogenetically informative. I wonder how this affects the analyses, and in particular the phylogenetic information content in data set B/C.

We respectfully disagree with the reviewer.

We have now performed new analyses in which we tested the use of a less stringent site selection approach. In these analyses the number of sites was increased (to 127,114 AA) – results are in the figure attached below. These analyses could only be run under ML so the trees reported are ML trees and should thus be compared to the ML trees in our MS (they use the same model). These trees confirm that our site selection strategy was not too stringent. This is because the ML trees from the larger dataset (127,114 sites) found topologies that are in excellent agreement with that of the ML trees in our MS (particularly supporting monophyletic Acari). However, it is to be noted that, when site number is increased to 127,114, an artefactual clade emerges that includes two sea spiders, two onychophorans and an amblypygid.

This clade is clearly artefactual as there is no evidence for either polyphyletic sea spiders, Onychophora or amblypygids, nor for a group composed of, sea spiders, onychophorans and amblypygids to be nested within Solifugae with bootstrap support of 100%!

An inspection of the quality of the transcriptomes used in our study (supplementary tables) clearly indicates that the five “rogue taxa” that constitute this “rogue clade” are those with the transcriptomes of poorer quality and the most missing data.

The effect of missing data in phylogenetic reconstruction is somewhat difficult to assess. However, there is abundant evidence that while missing data might not be problematic when randomly distributed in the dataset (as in the simulation of Philippe et al. 2004 MBE – H. Philippe personal communication), they negatively impact tree reconstruction when they are non-randomly distributed. Lemmon et al. (2009) presented evidence clearly demonstrating that non-randomly distributed missing data cause the inference artefactual branch lengths and topologies. Steel and Sanderson (2010) illustrated how non-randomly distributed missing data can affect “a first” form Data Decisiveness by completely precluding the possibility to discover specific tree topologies, and Roure et al. (2013) elegantly illustrated how non-randomly distributed missing data can magnify LBA artifacts. Finally, Dell’Ampio et al. (2014), in a paper co-authored by the reviewer, illustrated how non-randomly distributed missing data can contribute to the recovery of strongly supported artefactual clades, by affecting a “second form” (differently defined with reference to that of Steel and Sanderson 2010) of Data Decisiveness.

A Fuller investigation of why a rogue (clearly incorrect and strongly supported) clade emerges when we relax our site selection strategy is outside of the scope of the current MS, although it seems evident that this is a Missing-Data induced artifact, perhaps similar to that described by Dell’Ampio et al. (2014).

Notably, when the 5 gap-rich, rogue, taxa are excluded from the analyses (tree attached in the figure below (right)) a tree with the same topology of those in our MS is recovered, indicating that adding more sites did not affect the main relationships in our dataset, it only causes the emergence of a strongly supported missing-data-induced artifact.

In addition, it is worth pointing out that removal of more gapped sites (as in the trees presented in the MS) allowed for the clustering of all five rogue taxa as member of their correct order (sea spiders, and amblypygi) or phylum (Onychophora). As the monophyly of sea spiders, Amblypygi, and Onychophora is unambiguous, the difference between the tree derived using the 127,114 AA (figure below (left)) and the trees we presented in the MS clearly indicates that the removal of further gapped sites had an overall positive effect on the accuracy of the tree – it did not change relationships deeper in the tree while clearly improving relationships towards the tip of the tree (allowing recovery of monophyletic order and phyla).

These results indicate that our site-trimming strategy is sound and that adding more sites does not change our conclusions.

We would like to avoid inclusion of these new analyses in the MS as they do not add to (or alter) our main conclusions and make the paper more difficult to read.

Minor issues

1. The authors state that the genes in data set A are less saturated than those in data sets B, and figure 1A seems to support this. Now, my naive expectation would be the following: If I remove the data of all genes in data set B, which make up the less saturated data set A, then I would assume that the saturation would increase. However, figure 1A tells a different story. I have the feeling that this requires at least some attention (This is tightly linked to major issue 1).

We have added a statement in text to clarify this point as suggested by the reviewer.

To clarify, The key point in Fig 1a is that datasets B and C have effectively the same level of saturation, with dataset A being much less saturated. As the reviewer pointed out, dataset B only includes about 9.33% of the sites in dataset A and as these sites have not been selected with reference to their phylogenetic utility: they are not necessarily the least saturated sites in dataset A. The similar level of saturation between datasets B and C was to be expected (as only 9.33% of sites had to be removed from B to generate C). The observed difference in R-values between B and C clearly reflects random variation (they are not significantly different). This could be formally tested but it is very computationally intensive as it implies bootstrapping the data and performing ML and saturation plot analyses for each bootstrapped dataset, and it is outside the scope of the current MS, as what really is important to our argument is that Matrix A is much less saturated than Matrix B and this is already evident from Figure 1a.

2. Abstract – I suggest to remove the redundancy in the phrase “alternative, conflicting phylogenetic relationships”. Alternative relationships are always conflicting, and vice versa. Likewise, in the Results and Discussion section, I suggest to simplify the phrase ‘common set of shared genes’, which is equally redundant. **Done (we deleted “Alternative”)**

3. Page 3 – why is there a ‘~’ in front of the 40%? Probably it would be better to give the minimum number of species a gene has to be represented in rather than a percentage.

We deleted “~”, and we now explicitly state that orthologs must have been present in at least in 35% of taxa.

4. In data sets B/C, the authors have retained, prior to the Gblocks filtering, 3,982 genes that are represented in at least 40% of the 95 species. Given the Gblocks parameter settings, does a gene that is represented in say half of the 95 species a chance to survive the filtering? Somehow, and given that the resulting data matrix is

86.4% complete, I trust that this is not the case. I think this should be clarified. I am convinced that data sets B/C contain, after filtering, data from substantially fewer than 3,982 genes.

Unfortunately, this would be very difficult to test as Gblocks was applied to the concatenated alignments. However, to address the major comment of the reviewer (see above – that our trimming strategy was too stringent) we have now performed analyses based on a much more relaxed filtering strategy that find the same results. Accordingly, we do not think that this needs to be explicitly pointed out, as clearly our key results are also supported by datasets that include more than 3,982 genes, i.e. they are not an artefact caused by using relatively short alignments.

Note also that to make sure our results were not caused by the use of Gblocks, we used TrimAl to derive the trees attached to this reply, which clearly indicates that our key results are also independent of the software used to decide what sites to trim.

5. In the Results and Discussion section, the length of data sets A and B is specified, but not of data set C. Please add this information.

Done

6. I have some difficulties with the statement that CAT-GTR+G model is the optimal one, whereas LG and its variants are referred to as 'sub-optimal'. I think we are all aware that all models that we have at hand are 'sub-optimal', as we can be sure that they do not accurately model the true evolutionary process. In essence, from all models at hand for this analysis, CAT-GTR+G gives the best fit to the data. Still, when compared to an unknown 'true model' it can be (almost) as wrong as the LG model. This should be somehow reflected in the wording.

We agree with the reviewer that this was poorly expressed. We have changed suboptimal to "poorer fitting". Note that in the text we already said that CAT-GTR was best fitting, not optimal.

We changed all instances of "suboptimal" models that are now referred to as "poorer fitting".

7. Results and Discussion – The authors state that the Dayhoff recoded matrix confirms the support for a sister group relationship between horseshoe crab and Arachnida. Figure 1d seems to not support this statement since the split the authors refer to is not resolved. Instead we see a trifurcation.

The reviewer is correct, this was incorrectly formulated.

We have reformulated this as "does not reject the possibility that" (given that one of the possible resolutions of the polytomy is with horseshoe crabs as sister of Arachnida). We think that as now formulated it is accurate and precisely expresses the idea we would like to convey.

8. The section "Phylogenetic relationships within the Arachnida repeats the findings from the previous section when stating "all analyses recover monophyletic Acari". I leave it to the discretion of the authors to remove this redundancy.

We think this is the key message and we would like to reiterate it. No changes have been made.

9. The authors state that the Dayhoff recoded matrix A does not recover the grouping of Acari and Solifugae. They interpret this as a suggestion that this grouping might be an artefact. I would not support this interpretation for two reasons: (i) the split uniting Solifugae and Acari in Figure 1A is supported with a posterior probability of only 0.77 anyway. So, one should be careful with referring this to a clade anyway. (ii) The support for the placement of Solifugae in Figure 1d is only 0.64, so it is essentially unresolved. Thus, the recoded data matrix A seems to not contain sufficient signal to suggest anything with respect to whether or not Solifugae are sister to the Acari, or not.

We have reformulated this to clarify that support is low in Figure 1A, whereas in Figure 2 (which excludes unstable taxa - see caption of Figure 2) support goes up to PP=1.

In any case we agree with the reviewer and what we were (and are) trying to achieve in the text is to convey the idea that even the limited support observed for this clade might come from lineage-specific composition bias rather than real (but weak) phylogenetic signal. We wanted to clarify this point in the text as some readers might still consider this clade valid irrespective of the low support, given that there is some morphological support for it, and low PP indicates weak signal but not necessarily lack of accuracy, while the agreement (consilience) of morphology and molecules is potentially indicative of accuracy (irrespective of signal strength).

So we essentially agree with the reviewer but wanted to make the case a little stronger against Acari + Solifugae. We have revised the text to make all this clearer.

10. The authors write 'ambiguous support' when referring to the outcome of the ML analysis concerning the placement of Ricinulei and Solifugae. What is this 'ambiguous support' exactly, low bootstrap values? If so, why did they not use the same phrase in the context of low posterior probabilities?

We have changed this to low support (and we now talk of low support also for PP in the case of point 9 above).

11. What does it mean when "groupings are taxonomically familiar"?

The reviewer is correct that this was poorly formulated and the sentence has now been deleted.

12. Methods – what is an "effective set" of translated proteins?

We deleted "effective". What we meant was a set of translated proteins inferred from the transcriptomic sequences.

In summary, I hope that my comments help to further improve the manuscript.

Thanks. Your comments were very thorough and have helped us a lot to formulate a better paper.

Sincerely,

Ingo Ebersberger

Reviewer #3 (Remarks to the Author):

In their revised ms Lozano-Fernandez and co-authors have introduced number of changes following mine suggestions and those of other two reviewers. I find the revised version greatly improved. Some parts that were unclear in the original submission (e.g. model selection) are now better explained and I find the authors' reply to my comments and the corresponding changes in the manuscript satisfactory.

I felt that it may be a bit odd to use "rock record" to refer to the fossil record but that is just because fossil record is more commonly used. Both are correct indeed.

This is an interesting and important contribution to our knowledge of chelicerae relationships and particularly mites and ticks and I am happy to recommend publication of the ms.

Thanks. No changes required.